# The effect of health behavior interventions to manage Type 2 diabetes on the quality of life in low-and middle-income countries: A systematic review and meta-analysis

**Ashmita Karki** [1]*, **Corneel Vandelanotte**[1], **Saman Khalesi**[1], **Padam Dahal**[1], **Lal B. Rawal**[1,2]

**1** School of Health, Medical and Applied Sciences, Appleton Institute, Central Queensland University, Rockhampton, Australia, **2** Translational Health Research Institute (THRI), Western Sydney University, Sydney Australia

* Ashmita.karki@cqumail.com

## Abstract

### Background

Behavioral interventions targeted at managing Type 2 diabetes mellitus (T2DM) may have a positive effect on quality of life (QOL). Limited reviews have synthesized this effect in low- and middle-income countries (LMICs). This review and meta-analysis synthesised available evidence on the effect of behavioral interventions to manage T2DM on the QOL of people with T2DM in LMICs.

### Methods

Electronic databases PUBMED/MEDLINE, SCOPUS, CINAHL, Embase, Web of Science and PsycINFO were searched from May to June 2022. Studies published between January 2000 and May 2022, conducted in LMICs using randomized controlled trial design, using a health behavior intervention for T2DM management, and reporting QOL outcomes were included. Difference in QOL change scores between the intervention and control group was calculated as the standardized mean difference (SMD) of QOL scores observed between the intervention and control groups. Random-effects model was used for meta-analysis.

### Results

Of 6122 studies identified initially, 45 studies met the inclusion criteria (n = 8336). Of them, 31 involved diabetes self-management education and 14 included dietary and/or physical activity intervention. There was moderate quality evidence from the meta-analysis of mean QOL (n = 25) that health behavior intervention improved the QOL of people with T2DM (SMD = 1.62, 95%CI = 0.65–2.60 $I^2$ = 0.96, p = 0.001). However, no significant improvements were found for studies (n = 7) separately assessing the physical component summary (SMD = 0.76, 95%CI = -0.03–1.56 $I^2$ = 0.94, p = 0.060) and mental component summary (SMD = 0.43, 95%CI = -0.30–1.16 $I^2$ = 0.94, p = 0.249) scores. High heterogeneity and imprecise results across studies resulted in low to moderate quality of evidence.

**Data Availability Statement:** All relevant data are within the manuscript and its Supporting Information files.

**Funding:** The authors received no specific funding for this work.

**Competing interests:** The authors have declared that no competing interests exist.

## Conclusion

The findings suggest that health behavior interventions to manage T2DM may substantially improve the QOL of individuals with T2DM over short term. However, due to low to moderate quality of evidence, further research is required to corroborate our findings. Results of this review may guide future research and have policy implications for T2DM management in LMICs.

## Introduction

Type 2 diabetes mellitus (T2DM) accounts for around 90% of all cases of diabetes worldwide and is associated with poor health behaviors [1]. More than half a billion people are living with diabetes globally, 75% of whom live in low-and middle-income countries (LMICs) and this number is expected to reach 783 million by 2045 [2]. In addition, people living with T2DM are known to have lower quality of life (QOL) and more depressive symptomatology than healthy people [3]. Behavioral interventions such as diabetes self-management education (DSME), diet and/or physical activity interventions may have a significant impact on diabetes care and improving one's QOL [4]. In light of this, the aim of behavioral interventions has extended from just metabolic control to patient-oriented outcomes such as QOL [5,6]. This is particularly important because the use of QOL as an intervention outcome was found to empower people with T2DM to express their opinion on how the disease and the intervention affects their health and to take ownership of their care [7]. Furthermore, improved QOL is significantly associated with improved diabetes self-management as shown by a review of studies mostly conducted in high income countries (HICs) [5].

As a key element in chronic care, DSME has been found effective in improving metabolic control [8], behavioral outcomes [9], and QOL [10,11]. Previous reviews from LMICs have shown the effectiveness of DSME on metabolic parameters [6,12], however, outcomes in relation to QOL are inconclusive due to limited data availability on QOL outcomes [13]. This paucity of data due to the complex construct of QOL combined with likely measurement error makes the assessment of QOL quite challenging [14]. A review of twelve studies, conducted in four LMICs and two HICs indicated that DSME interventions were effective in improving clinical parameters as well as patient-reported outcomes such as self-management behavior, QOL and self-efficacy [15]. Similarly, another review of eight studies, conducted in five HICs and three LMICs demonstrated QOL improvement in 3 out of the 5 randomized controlled trials (RCTs) that reported change in QOL measures [16]. However, these reviews only included studies with DSME components, and excluded studies solely focussing on other health behavior interventions such as structured dietary and/or exercise programs. In addition, these reviews have not been synthesized to produce a pooled estimate of the effect size, which is critical to strengthen the existing evidence base. Moreover, there is a dearth of reviews exclusively focusing on LMICs.

Similarly, dietary and physical activity interventions have shown to reduce glycated haemoglobin (HbA1c) [17,18], including a recent meta-analysis showing that moderate to vigorous intensity aerobic physical activity for thirty minutes per week is significantly associated with HbA1c reduction [19]. However, reviews and meta-analyses have mainly reported on the metabolic markers in terms of outcome, rather than on psychological and QOL outcomes [20], due to the limited data available and challenges in calculating effect sizes due to using different QOL measurement tools [6]. Moreover, these studies have largely been conducted in high income countries, creating the need for reviews focusing on T2DM in LMICs.

In sum, studies worldwide have shown that health behavior interventions are effective in optimising glycemic control and diabetes management [12,19]. However, despite QOL being an important measure in T2DM management, the synthesis of evidence on the effect of these interventions on the QOL of people with T2DM in LMICs is limited. Furthermore, to our knowledge, no systematic review and meta-analysis has ever been conducted to synthesize available evidence on the effect of health behavior interventions to manage T2DM on the QOL of people in LMICs [12]. The evidence generated from this study can be used to help guide future research and clinical practice and may have policy implications in the management of T2DM in LMICs.

## Methods

### Reporting and registration of the review

This systematic review adhered to the Preferred Reporting Items for Systematic Reviews and Meta-Analysis (PRISMA) guideline [21] and followed the Joanna Briggs Institute (JBI) systematic review methodology [22]. The review protocol was registered in the PROSPERO International Prospective Register of systematic reviews (Registration ID CRD42022323184) on 05 July 2022. Ethical approval was not required for conducting this systematic review and meta-analysis.

### Data sources and searches

Data were searched from 26 May to 1 June 2022. Online databases PUBMED/MEDLINE, SCOPUS, CINAHL, Embase, Web of Science and PsycINFO were searched using the pre-defined search terms, which were based on the PICOS (population, intervention, comparison, outcome, and study design) framework. The population group was people with T2DM; intervention was any health behavioural intervention targeted at T2DM management; comparison group was usual care, waitlist or attentional control group; and study design was randomized controlled trial. A manual search of reference lists of included studies was also conducted to identify additional studies that met the inclusion criteria. A detailed list of search terms was developed using the combinations of mainly four key words, "Type 2 diabetes mellitus", "Quality of life", "Health behavior interventions" and "Randomized controlled trials". The search terms can be seen in S1 Table. Studies were searched regardless of the location/region, then segregated by income region, rather than using the search terms for LMICs, as not all studies conducted in LMICs might have identified themselves as such. A university research librarian was consulted to optimise the search strategy.

### Inclusion and exclusion criteria

Only studies with an RCT design were included in this review to ensure that the evidence provided on the effectiveness of interventions was robust. When multiple articles from the same study/studies were found, only the article that reported the most relevant QOL information/outcome was included. All types of health behavior interventions were included regardless of the setting. Only studies conducted between January 2000 and May 2022 and published in English language were included in the review, as examining QOL in individuals with T2DM is a relatively new concept. Studies were included if they 1) examined any behavioral or educational interventions targeted at improving T2DM management among people with new or established diagnosis of T2DM; 2) reported QOL outcomes using validated QOL measures, both pre- and post-intervention, as a primary or secondary outcome; 3) were conducted within an RCT design; and 4) were conducted in LMICs, as defined by the World Bank [23].

Studies were excluded if they 1) were published in languages other than English; 2) had therapeutic or pharmacological intervention strategies; 3) had mixed study population of Type 1 and 2 diabetes with no separate data reported for T2DM population; 4) were observational studies or reported in reviews, editorials, theses, books, short communication; or 5) presented inadequate or unclear QOL data.

## Study selection

The articles retrieved from the search were exported to COVIDENCE [24], a web-based software developed to streamline systematic review process on 2 June 2022. After removing duplicates, title and abstract of the identified studies were independently screened by two reviewers (AK and PD) based on pre-defined inclusion criteria. Full texts of retrieved studies were then screened by two reviewers (AK and PD) independently. Any disagreements were resolved through consultation with other team members. All studies excluded at the full-text stage were recorded with detailed reasons of exclusion during screening. The study selection was completed on 22 July 2022. A PRISMA flow diagram is shown in Fig 1.

## Outcome measures

The primary outcome measure of this review was difference in QOL change scores between the intervention and control groups. The secondary outcomes were HbA1c, fasting blood glucose (FBG), diabetes self-management behavior, and anthropometric measures such as BMI and weight.

## Data extraction

Author name, publication year, study objective, country, setting, study design, demographic characteristics such as age, gender, etc., intervention description, control details, duration of intervention and follow up, QOL measurement tools, and primary and secondary intervention outcomes of interest were extracted using a template by one author (AK) and their accuracy were checked by the second author (PD) [25] from 23 July to 25 August 2022.

## Quality assessment

The 13-item JBI Critical Appraisal Checklist for Randomized Controlled Trials was used to assess methodological quality of the selected studies [26]. Two reviewers (AK and PD) independently assessed the risk of bias. Any disagreements were discussed with the team members until consensus was reached. Each item in the JBI checklist was scored one if they fulfilled the criteria for that item and scored zero if they did not fulfil the criteria. For example, if a study reported blinding of outcome assessors, then the study was scored '1' for the item about blinding of outcome assessors, whereas, if the study did not report blinding of outcome assessors, then the study was scored '0'. Summary scores were obtained for each selected studies by adding the item-specific scores. The quality of the studies was then rated as good ($\geq$8), fair (6–7), or poor ($\leq$5) based on the summary scores [27]. In addition, the quality of evidence across RCTs included in the meta-analysis was assessed by two reviewers (AK and PD) using the Grading of Recommendations, Assessment, Development and Evaluation (GRADE) approach, and rated as high, moderate, low or very low [28]. Since all included studies were RCTs, the rating began with a high-certainty rating. Then, the quality was upgraded or downgraded based on the following criteria: i) quality across the studies as determined by the JBI Critical Appraisal Checklist for Randomized Controlled Trials; ii) inconsistency/heterogeneity level; iii) indirectness of evidence; iv) imprecise results (wide 95% CI i.e., > 0.8 SMD); and v)

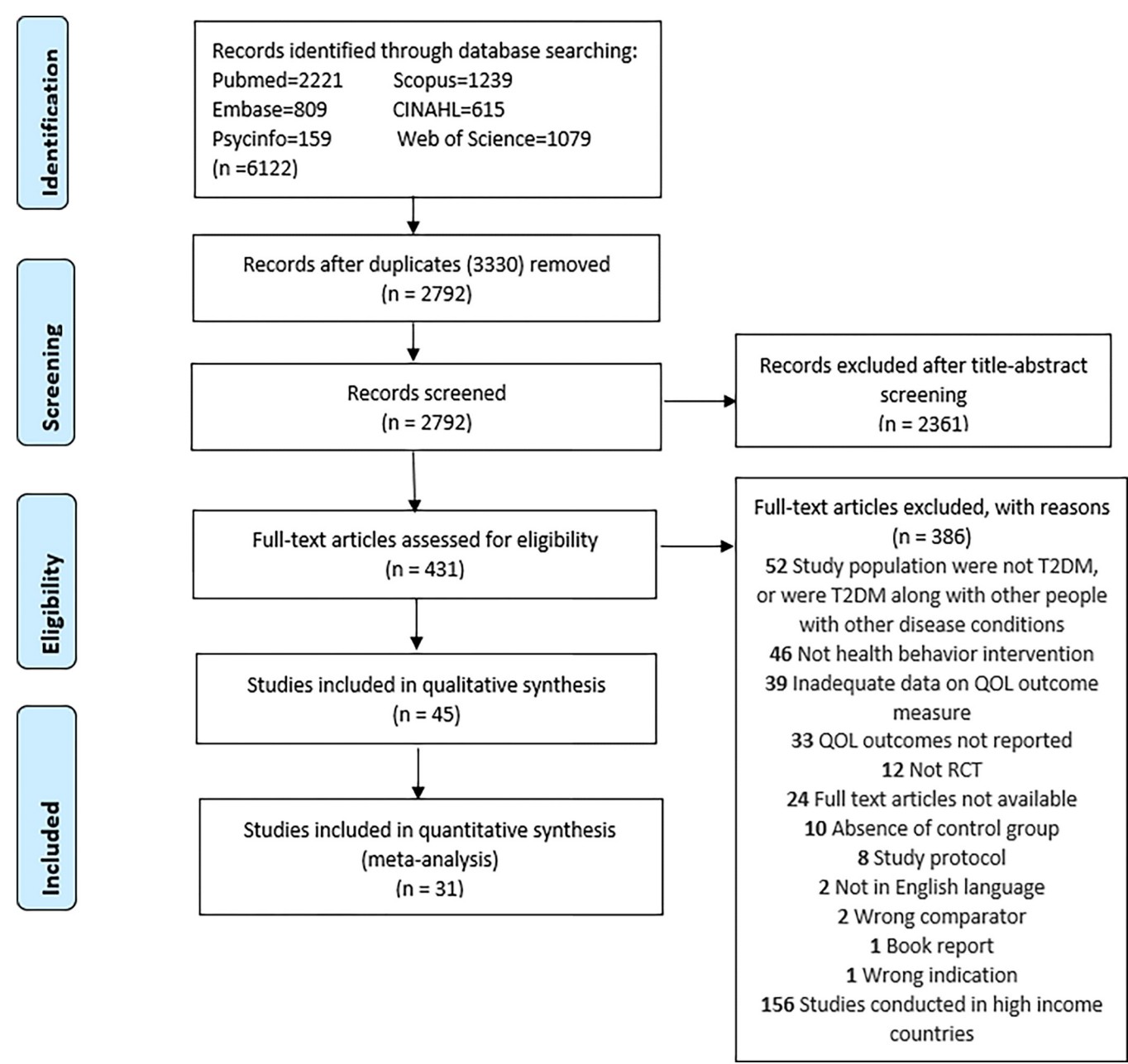

**Fig 1. Preferred Reporting Items for Systematic Reviews and Meta-analyses (PRISMA) flow diagram.**

publication bias (visual inspection of funnel plot) [29]. If the majority of the RCTs scored ≤ 5 on the JBI scale, the certainty of evidence was downgraded two places whereas, if the effect size was large (SMD ≥ 0.8), then the evidence was upgraded one place [29].

## Data synthesis and analysis

Given the number of different scales used to measure QOL scores, the effect of T2DM health behavior interventions was calculated as the standardized mean difference (SMD) of QOL change scores observed between the intervention and control groups. The Cochrane

Handbook for Systematic Review of Interventions guidelines [30] was followed to calculate mean and standard deviation (SD) of change from baseline in the intervention and control groups. Given that none of the studies included in the meta-analysis reported the absolute mean and SD of change, a correlation coefficient (r) of 0.6 was assumed [30] to calculate the SD of change using the following formula:

$$SD_{Change} = \sqrt{SD_{Baseline}^2 + SD_{Final}^2 - (2 \times r \times SD_{Baseline} \times SD_{Final})}$$

QOL score was reported as overall QOL score in majority of the studies. Physical component summary (PCS) and/or mental component summary (MCS) scores were also reported in some studies that used Short Form (SF) questionnaires to measure QOL. Therefore, three separate meta-analyses were performed to investigate the effect of health behavior intervention on overall, PCS, or MCS scores. The magnitude of effect of the behavior interventions on QOL was defined according to Cohen recommendation on Hedge's g levels and interpreted as small (g ≤ 0.2), medium (g = 0.2 to 0.5), and high (g≥ 0.8) effect [31]. Heterogeneity was assessed using Cochran's $I^2$ index with values <40, 40–75, and >75% corresponded to low, moderate and high heterogeneity respectively [30]. Similarly, funnel plots were used to assess publication bias. Given the heterogeneous nature of the interventions and QOL measurement tools used, a random-effects model was applied to assess the mean and variability of effect sizes across studies. For studies with QOL measurements at multiple time-points, only the measurements at baseline and following intervention completion (post-intervention) were included in the meta-analysis to maintain consistency across studies using a similar timepoint and improve homogeneity in data extraction.

Subgroup analyses were performed to investigate differences in the effect of behavior change intervention on QOL scores based on intervention type (DSME, diet and/or physical activity interventions), settings (hospital/diabetes clinic, community health centre, home/web-based), intervention duration (less than 12 weeks, 12 weeks or longer), QOL scale (generic, disease-specific), and methodology quality (good, fair, poor). Sub-group analyses based on QOL scale and methodology quality were added post hoc. Meta-regression analyses were performed to determine the linear relationship between duration as a continuous covariate and the effect size. Meta analyses using different correlation coefficients (r = 0.8 and 0.4) were also conducted to compare the direction of effect and heterogeneity with the assumed coefficient (r = 0.6). All statistical analyses were performed using RStudio software, version 1.3.1073 [32]. The packages 'meta' [33] and 'dmetar' [34] were used for the analysis (R codes used in this study are available in the appendix as S1 Fig). All data are presented as standardized mean ± confidence interval (CI).

## Results

### Study selection

A total of 6122 studies were initially identified through database searching. After removing 3330 duplicates, 2792 studies were reviewed by title and abstract, of which, 431 progressed to full text review. Altogether, 386 studies were excluded after full text review. The list of excluded articles is presented in the appendix as S7 Table. Finally, 45 articles were included in the review, and 31 progressed to meta-analysis as illustrated in Fig 1.

### Study characteristics

Twenty-four studies were conducted in lower middle-income countries [8,10,11,35–55] and 21 were conducted in upper middle-income countries [9,56–75]. The study duration ranged from 4 weeks [41] to 152 weeks [43], with more than half of studies exceeding a six-month

study period. The total sample size was 8336, with individual sample sizes ranging from 30 [44] to 1570 [56]; 27 out of 45 studies had a sample size greater than 100. The mean age of the study participants ranged from 31 to 74 years. The percentage of female participants was higher than that of male participants in two-thirds of the studies. All studies included in the review reported a QOL measure, 27 reported HbA1c, 12 investigated changes in self-management behavior, and 3 examined changes in medication adherence. There was no dropout in the intervention group of 10 studies [11,35,41,45,46,48,49,64,75,76], while the highest intervention group dropout reported was 54.9% [67]. Similarly, there was no dropout in the control group of 11 studies [11,35,41,44,45,49,55,64,74–76], while the highest control group dropout reported was 44.8% [57]. Detailed study characteristics are presented in Table 1.

## Health behavior intervention characteristics

Thirty-one out of 45 studies involved DSME interventions, the education content of which focused on physical activity and/or diet (n = 23), medication advice (n = 20), glucose monitoring (n = 19), foot care (n = 17), problem solving and stress reduction (n = 13). Of the remaining 14 studies, 10 were structured exercise programs [35,37,40,44,46,48,55,63,67,75], two were structured dietary programs [52,64], and two were the combination of structured exercise and DSME programs [39,50].

The majority of the interventions (28) were delivered by health care professionals (HCP) [8,9,41,43,50,54,57–60,62,65,68,70–73], followed by allied health professionals (AHP)/exercise trainers [37,44,46,47,52,55,63,67,75] and peer-supporter or lay people trained to deliver the intervention [45,48,49,56,61]. There was no mention of who delivered the interventions in eight studies [11,35,36,38,53,64,69,74].

The intervention settings varied from community health centres [38,39,56,59,61,65,66,68, 72,74], hospital/diabetes clinics [8–11,36,37,40–43,45–47,49,50,52–54,57,58,60,62–64,67,70,71,73] to home/web-based [48,55]. The intervention duration varied from four weeks [41] to 104 weeks [51], with majority of the studies delivering the intervention for three months or more. The median duration of the interventions was 18 weeks.

Fifteen studies [9,37,43,46,48,52,56–58,63,64,71,74,75,77] reported using various versions of SF questionnaires (SF-12, SF-20, SF-36) as a QOL measurement tool. The remaining studies used other generic (e.g., the World Health Organisation QOL questionnaire) and diabetes-specific (e.g., Diabetes-39) questionnaires, as summarized in S2 Table. Similarly, mean values for the overall QOL or domain specific QOL were extracted along with the values of secondary outcome measures (S3 Table).

## Quality assessment

An overview of the quality assessment of the studies is presented in the appendix as S4 Table. The overall quality of the studies ranged from "fair" to "good" after obtaining a summary score of at least six in the quality assessment. No study was rated as poor quality as none obtained a summary score of five or less. Allocation of study participants to intervention and control groups wasn't concealed in three studies and not clearly indicated in 23 studies. Only two studies clearly specified blinding of participants, five stated blinding of those delivering the intervention, and 13 specified blinding of outcome assessors. Use of appropriate trial design and statistical analysis methodology was strong in all studies. Thirty-eight studies provided information on participants' attrition with reasons for dropouts and withdrawals. In 23 studies, the data were analysed following the intention-to-treat principle.

The quality of evidence across studies included in the meta-analyses, assessed using the GRADE approach, is presented in the appendix as S5 Table, and the description is provided

**Table 1. General characteristics of the studies included in the review sorted by author name alphabetically.**

| Author | Year | Objective of the study | Study country | Country classification | Intervention type | Intervention detail | Control detail | Study duration (weeks) | Sample size (I/C) | Study drop-out (I/C) % | Age Mean (SD) | Gender (I % Male/C % Male) | Outcome measures of interest | QOL tool |
|---|---|---|---|---|---|---|---|---|---|---|---|---|---|---|
| Abraham [47] | 2020 | to determine the effect of a cognitive educational intervention on HbA1c, QOL, selfcare and psychosocial measures | India | LMIC | SME/CBT | Educational intervention based on cognitive behavioral principles | Usual routine medical care | 22 | 40/40 | 25.0/37.5 | I: 50.4 (9.3) C: 50.6 (8.3) | 44/42 | QOL, HbA1c, self-care | DQOL |
| Akinci [67] | 2018 | to analyse the effects of supervised aerobic and resistance exercise and web-based exercise in people with T2DM | Turkey | UMIC | Structured exercise | 1. Supervised aerobic and resistance exercise 2. Web-based exercise performed with the help of exercise videos on aerobic and resistance exercises | Brochure on importance and benefits of physical activity for people with diabetes | 8 | 22/22/22 | 9.0/54.5/9.0 | I1: 53.6 (6.0) I2: 50.2 (6.5) C: 53.6 (6.7) | 18/33/38 | QOL, HbA1c, physical activity, FBG, BMI | EQ-5D-5L |
| Anderson [51] | 2009 | to determine the effectiveness of an empowerment-based self-management consultant in improving the glycemic and QOL outcomes in people with T2DM | India | LMIC | SME | Nurse and dietitian-led diabetes self-management consultation based on empowerment approach | Mailed results of their metabolic assessments only | 104 | 156/154 | 21.8/17.5 | I: 55.5 (11.3) C: 55.7 (11.5) | 44/39 | QOL, HbA1c | PAID |
| Arora [44] | 2009 | to examine the effects of progressive resistance training compared with aerobic exercise on glycemic and well-being outcomes in people with T2DM | India | LMIC | Structured exercise | 1. Supervised progressive resistance training 2. Supervised aerobic exercise | Usual care with no exercise training but continued with their medication and nutritional regimen as before | 8 | 10/10/10 | 5.0/0 | I1: 49.6 (5.2) I2: 52.2 (9.3) C: 58.4 (1.8) | 40/60/60 | Well-being, HbA1c, BMI | WBQ |
| Arovah [55] | 2018 | to determine the impact of a pedometer-based walking program based on social cognitive theory in people with T2DM | Indonesia | LMIC | Structured exercise | Social cognitive theory based physical activity intervention added to the provision of pedometers, also known as Walking with diabetes (WW-DIAB) program | Pedometer and log sheet for recording steps only, which was not the focus of the research | 24 | 21/22 | 4.7/0 | I: 65.1 (5.2) C: 65.9 (6.5) | 68/36 | QOL, HbA1c, physical activity | EQ-5D |

(Continued)

**Table 1.** (Continued)

| Author | Year | Objective of the study | Study country | Country classification | Intervention type | Intervention detail | Control detail | Study duration (weeks) | Sample size (I/C) | Study drop-out (I/C) % | Age Mean (SD) | Gender (I % Male/C % Male) | Outcome measures of interest | QOL tool |
|---|---|---|---|---|---|---|---|---|---|---|---|---|---|---|
| Azami [8] | 2018 | to determine the effectiveness of a nurse-led diabetes self-management education on HbA1C. | Iran | LMIC | SME | Self-efficacy and motivational interviewing-based SME and follow-up telephone calls | Usual diabetes routine care | 26 | 71/71 | 2.8/5.6 | I: 55.1 (10.2) C: 53.5 (10.9) | 32/37 | HbA1c, weight, BMI, QOL | WHOQOL-Bref |
| Browning [66] | 2016 | to determine the effect of a coach-led educational intervention, based on motivational interviewing, in the glycemic and psychosocial outcomes of people with T2DM | China | UMIC | SME | Telephone and face-to-face health coaching based on motivational interviewing | Usual care from Community health stations as per the Chinese Guideline for Diabetes Prevention and Management | 104 | 385/345 | 15.5/12.4 | I: 63.7 (7.6) C: 64.0 (9.0) | 49/46 | QOL, BMI, HbA1c, self-care activities | WHOQOL-Bref |
| Butt [60] | 2015 | to analyze the impact of a pharmacist-led diabetes education program on HbA1c, medication compliance and QOL | Malaysia | UMIC | SME | Diabetes management education programme, called Patient Education by Pharmacist Programme | Usual care at a medical centre which consisted of patient-physician meeting | 26 | 37/36 | 10.8/8.3 | I: 57.4 (7.2) C: 57.1 (10.8) | 39/42 | HbA1c, BMI, QOL | EQ-5D-3L |
| Cani [70] | 2015 | to analyse the impact of a pharmacist-led program in diabetes self-mangement among people with T2DM | Brazil | UMIC | SME | Individualized pharmacotherapeutic care plan incorporating diabetes education | Usual standard care | 26 | 37/41 | 8.1/12.1 | I: 61.91 (9.6) C: 61.58 (8.1) | 38/39 | QOL, HbA1c, medication adherence | DQOL |
| Castillo-Hernandez [61] | 2020 | to explore if peer support in addition to a diabetes management education improves HbA1c and QOL outcomes | Mexico | UMIC | SME | Peer support SME for T2DM patients | SME | 35 | 29/29 | 3.4/10.3 | I: 59.0 (9.4) C: 56.0 (10.3) | 7/0 | HbA1c, QOL, BMI, QOL, self-care behaviors | Diabetes-39 |
| Chaveepojnkamjorn [65] | 2009 | to assess the effect of a self-help group program on the QOL of people with T2DM | Thailand | UMIC | SME | Group education session and active learning for diabetes management | Usual care at a health centre | 30 | 80/84 | 8.7/13.1 | I: 48.9 (6.9) C: 49.1 (7.3) | 22/23 | QOL | WHOQOL-Bref |

*(Continued)*

**Table 1.** (Continued)

| Author | Year | Objective of the study | Study country | Country classification | Intervention type | Intervention detail | Control detail | Study duration (weeks) | Sample size (I/C) | Study drop-out (I/C) % | Age Mean (SD) | Gender (I % Male/C % Male) | Outcome measures of interest | QOL tool |
|---|---|---|---|---|---|---|---|---|---|---|---|---|---|---|
| Cheng [62] | 2019 | to analyse the effects of a patient-centered empowerment based education approach on psychological outcomes in people with T2DM | China | UMIC | SME | Empowerment-based education intervention | Attentional control, routine care with two general health education classes and social calls post-discharge | 78 | 121/121 | 15.7/18.1 | I: 56.1 (10.7) C: 53.9 (13.0) | 77/71 | QOL | ADDQOL |
| Dede [75] | 2015 | to study the impact of a moderate intensity exercise training on the QOL of people with T2DM | Turkey | UMIC | Structured exercise | Supervised moderate intensity aerobic exercise training | Normal daily activities without additional guided physical activities | 12 | 30/30 | 0/0 | I: 52.5 (7.5) C: 55.5 (8.4) | 50/47 | QOL, BMI | SF-36 |
| Ebrahimi [10] | 2018 | to determine the impact of family-based education on diabetes management in QOL outcomes of people with T2DM | Iran | LMIC | SME | Diabetes management education to T2DM patients and their families | Usual standard care | 12 | 40/40 | 5.0/5.0 | I: 58.6 (7.7) C: 53.5 (9.2) | 34/24 | QOL | DCQOL |
| Jaipakdee [68] | 2015 | to determine the effect of a self-management support program aided with computer-assisted instruction in people with T2DM | Thailand | UMIC | SME | Nurse-led diabetes support program equipped with a computer-aided instruction, designed in the RE-AIM framework | Usual routine medical care | 26 | 203/200 | 4.4/8.0 | I: 61.1 (9.6) C: 61.5 (9.7) | 24/23 | QOL, HbA1c, health behavior score | DQOL |
| Jamshidpour [46] | 2020 | to assess the impact of combined moderate intensity aerobic and resistance training in people with T2DM undergoing hemodialysis | Iran | LMIC | Structured exercise | Combined aerobic and resistance exercise performed at moderate intensity (11-15/20 on the Borg scale) during hemodialysis treatment | Usual unmonitored and unrecorded physical activity | 8 | 15/15 | 0/13.3 | I: 64.9 (7.8) C: 58.5 (11.9) | 80/61 | QOL | SF-36 |

(Continued)

**Table 1.** (Continued)

| Author | Year | Objective of the study | Study country | Country classification | Intervention type | Intervention detail | Control detail | Study duration (weeks) | Sample size (I/C) | Study drop-out (I/C) % | Age Mean (SD) | Gender (I % Male/C % Male) | Outcome measures of interest | QOL tool |
|---|---|---|---|---|---|---|---|---|---|---|---|---|---|---|
| Kong [72] | 2019 | to assess the effect of a chronic-care model-based intervention on T2DM management | China | UMIC | SME | Comprehensive chronic care model to improve lifestyle behaviors | Conventional care at the community health service centre | 39 | 150/150 | 10.6/17.3 | I: 69.1 (10.5) C: 71.5 (8.8) | 42/44 | QOL, BMI | SF-36 |
| Lyu [57] | 2021 | to develop and examine the effects of a web-based transitional care program on glycemic control and quality of life in Chinese population with T2DM | China | UMIC | SME | Web-based courses on knowledge of diabetes and self-management | Usual care; routine discharge education and a diabetes knowledge manual consisting of general information on diabetes before discharge | 13 | 54/52 | 6.9/10.3 | I: 60.0 (10.0) C: 61.7 (10.5) | 52/44 | HbA1c, QOL, treatment adherence | SF-36 |
| Maharaj [37] | 2015 | to analyze the effects of rebound exercise and treadmill walking on the QOL of people with T2DM | Nigeria | LMIC | Structured exercise | 1. Supervised, structured moderate intensity rebound exercise 2. Treadmill walking for patients with T2DM | Routine medication and counselling; health magazines to read | 12 | 50/50/50 | 24.0/12.0 | I 1: 38.7 (5.6) I 2: 40.8 (5.5) C: 40.0 (6.1) | 59/51/52 | QOL | SF-36 |
| Mash [56] | 2014 | to assess the effectiveness of group education in T2DM management in under-resourced settings in South Africa | South Africa | UMIC | SME | Group education based on diabetes management | Usual education at the health centre that consisted of ad hoc educational talks | 52 | 710/860 | 44.9/44.8 | I: 55.8 (11.5) C: 56.4 (11.6) | 28/24 | HbA1c, weight, QOL, self-care activities | SF-20 |
| Mohammadi [36] | 2018 | to assess the impact of self-efficacy education based on Health Belief Model (HBM) in type 2 diabetes patients | Iran | LMIC | SME | Self-efficacy based diabetes education sessions | Only conventional dietary counselling | 36 | 120/120 | 8.3/8.3 | I: 51.2 (6.2) C: 51.4 (6.0) | - | HbA1c, FBG, weight, BMI, QOL | DQOL |

*(Continued)*

**Table 1.** (Continued)

| Author | Year | Objective of the study | Study country | Country classification | Intervention type | Intervention detail | Control detail | Study duration (weeks) | Sample size (I/C) | Study drop-out (I/C) % | Age Mean (SD) | Gender (I % Male/C % Male) | Outcome measures of interest | QOL tool |
|---|---|---|---|---|---|---|---|---|---|---|---|---|---|---|
| Nazir [54] | 2020 | to examine the effectiveness of an educational intervention in a pharmacist led, medication management program targetted for people with T2DM | Pakistan | LMIC | SME, medication adherence | Pharmaceutical care/ patient education through the medication therapy management program | Usual routine medical care at a hospital | 13 | 196/196 | 17.8/16.3 | - | 58/56 | QOL, HbA1c | EQ-5D |
| Nouripour [52] | 2021 | to examine the effect of high-protein versus high-carbohydrate evening diet on the quality of life of people with T2DM | Iran | LMIC | Structured dietary | high-carbohydrate versus high-protein intake during evening meal | Standard evening meal | 10 | 31/29/36 | 12.9/27.5/8.3 | I1: 54.0 (6.3) I2: 51.7 (8.2) C: 56.1 (7.2) | 52/48/44 | QOL | SF-36 |
| Peimani [49] | 2018 | to assess the effectiveness of a peer support education program targetted at improving self-care behaviors and QOL in people with T2DM | Iran | LMIC | SME | Peer support group meetings focused on experience sharing and problem solving in diabetes self-care and management | Usual clinic education | 26 | 100/100 | 0/0 | I: 59.0 (11.3) C: 58.8 (11.7) | 53/51 | QOL, HbA1c, BMI, self-management | SWED-QUAL |
| Rasoul [11] | 2019 | to assess the effect of self-management education imparted through weblogs on the QOL of people with T2DM | Iran | LMIC | SME | Self-management education through weblog | Routine care at a diabetes centre | 22 | 49/49 | 0/0 | I: 31.4 (5.3) C: 32.9 (4.4) | 53 | QOL, BMI | DQOL |
| Rias [48] | 2020 | to compare the effects of regular walking, consumption of alkaline electrolysed water, and their combined effect in people with T2DM | Indonesia | LMIC | Structured exercise | 1. Regular walking for at least 150 min/week 2. Drink 2 L/day of alkaline electrolysed water 3. Drink 2 L/ day of alkaline electrolysed water and regular walk for at least 150 min/week | Continued with their habitual diet and physical activity | 8 | 20/20/20/21 | 0/0/0/4.7 | I1: 54.7 (4.9) I2: 57.5 (5.5) I3: 56.2 (4.9) C: 55.7 (4.9) | 35/45/35/38 | QOL, FBG | SF-36 |

*(Continued)*

**Table 1.** (Continued)

| Author | Year | Objective of the study | Study country | Country classification | Intervention type | Intervention detail | Control detail | Study duration (weeks) | Sample size (I/C) | Study drop-out (I/C) % | Age Mean (SD) | Gender (I % Male/C % Male) | Outcome measures of interest | QOL tool |
|---|---|---|---|---|---|---|---|---|---|---|---|---|---|---|
| Rondhianto [38] | 2018 | to investigate the effect of diabetes education based on health belief model on psychosocial and glycemic outcomes | Indonesia | LMIC | SME | Education program based on the health belief model | Usual care | 18 | 60/60 | Unclear | I: 57.5 (6.8) C: 57.7 (5.7) | 30/43 | HbA1c, QOL, self-care behavior | Diabetes QOL scale |
| Safavi [73] | 2011 | to assess the effectiveness of an education program in improving the QOL and self-esteem in people with T2DM | Iran | UMIC | SME | Diabetes management education focussed on quality of life improvement | Waitlist control; handouts and the same QOL education programs after post-intervention data collection | 60 | 61/62 | Unclear | I>C (p = 0.016) | 49/48 | QOL, BMI | Farrell and Grant quality of life questionnaire |
| Saghaee [41] | 2020 | to assess the impact of a culture and theory based self-management education on self care activity and QOL of people with T2DM | Iran | LMIC | SME | Culture-oriented, theory and evidence-based, education workshops based on management of diabetes and reducing risk of complications | Non-interactive routine diabetes education | 4 | 17/17 | 0/0 | I: 66.3 (6.4) C: 69.1 (7.8) | 53/65 | QOL, self-care behavior | Diabetes QOL Brief Clinical Inventory (DQOL-BCI) |
| Sekhar [43] | 2019 | to study the effect of patient education on patient with diabetic foot ulcer | India | LMIC | SME | Patient counselling and education on foot care measures using patient information leaflets | Usual care | 152 | 210* | 35.7# | I: 58.6 (7.9) C: 60.3 (8.4) | 79/74 | QOL | SF-36 |
| Shahsavari [45] | 2021 | to evaluate the effect of peer support on the quality of life of people with T2DM | Iran | LMIC | SME | Peer support intervention on diabetes self-care and telephone follow up for T2DM patients | Usual routine medical care | 13 | 40/40 | 0/0 | I: 53.6 (14.3) C: 54.5 (12.9) | 43/40 | QOL | DQOL-BCI |
| Shenoy [35] | 2009 | to analyze the effects of aerobic walking on HbA1c, blood glucose and well-being in patients with type 2 diabetes | India | LMIC | Structured exercise | Supervised walking program using heart rate monitor and pedometer | No exercise training; continued with their medication as before | 8 | 20/20 | 0/0 | I: 53.1 (4.4) C: 51.0 (5.4) | 75/70 | HbA1c, BMI, General well-being | WBQ |

*(Continued)*

**Table 1.** (Continued)

| Author | Year | Objective of the study | Study country | Country classification | Intervention type | Intervention detail | Control detail | Study duration (weeks) | Sample size (I/C) | Study drop-out (I/C) % | Age Mean (SD) | Gender (I % Male/C % Male) | Outcome measures of interest | QOL tool |
|---|---|---|---|---|---|---|---|---|---|---|---|---|---|---|
| Shi [71] | 2018 | to determine the effect of an integrative health education based on Chinese medicine and Western medicine therapy in the glycemic and QOL outcomes in people with T2DM | China | UMIC | SME | Integrative health education including Western medicine education about diabetes health and therapeutic Chinese method such as Chinese dietary therapy and food choices | Usual education with handbooks of diabetes | 52 | 129/127 | 6.9/5.5 | I: 52.95 (14.0) C: 55.3 (13.4) | 46/46 | QOL, BMI | SF-36 |
| Singh [40] | 2020 | to investigate the effect of yoga and exercise on glycemic control, QOL outcome, and exercise self-efficacy | India | LMIC | Structured exercise | Supervised yoga sessions for 2 weeks, followed by home practice for 3 months | Usual care and general information on diabetes | 14 | 112/115 | 10.3/13.9 | I: 50.3 (9.1) C: 49.4 (8.7) | 41/49 | QOL, HbA1c | QOLID |
| Sreedevi [39] | 2017 | to assess the effects of yoga and peer support intervention on women with T2DM | India | LMIC | Structured exercise and peer-led SME | 1. Instructor driven yoga sessions 2. Peer support intervention by peer mentors identified from the community | Usual standard of care | 12 | 41/42/41 | 22.8/14.6 | I 1: 52.0 (7.4) I 2: 51.9 (8.3) C: 51.9 (6.6) | 0/0/0 | QOL, HbA1c, FPG, medication adherence | WHOQOL-Bref |
| Sunil [53] | 2020 | to assess the impact of a mobile application focusing on lifestyle change and medication management in the QOL of people with T2DM | India | LMIC | SME | Smartphone application named 'Diaguru' for lifestyle modification and medication management | Usual routine medical care | 26 | 150/150 | 0/0 | I: 55.7 (10.5) C: 73.6 (11.3) | 60/60 | QOL | WHOQOL-Bref |
| Tapehsari [50] | 2020 | to study the effect of Physical Activity Package (PAP) conducted with the help of an exercise prescription on the quality of life of people with T2DM | Iran | LMIC | Structured exercise with SME | Physical activity package program run with the help of exercise prescriptions by physicians | Usual general lifestyle education | 12 | 50/50 | 6.0/4.0 | I: 45.9 (6.9) C: 46.6 (5.3) | 19/15 | QOL, FBG | WHOQOL-Bref |

(Continued)

**Table 1.** (Continued)

| Author | Year | Objective of the study | Study country | Country classification | Intervention type | Intervention detail | Control detail | Study duration (weeks) | Sample size (I/C) | Study drop-out (I/C) % | Age Mean (SD) | Gender (I % Male/C % Male) | Outcome measures of interest | QOL tool |
|---|---|---|---|---|---|---|---|---|---|---|---|---|---|---|
| Torabizadeh [42] | 2018 | to determine the impact of problem-solving technique on glycemic outcome and QOL of cognitively impaired people with T2DM | Iran | LMIC | SME/CBT | Empowerment-based education intervention with problem-solving approach | Usual routine classes in the clinic | 19 | 50/50 | 2.0/6.0 | I: 52.6 (7.0) C: 50.7 (7.5) | 27/21 | QOL, HbA1c, FBG, self-care behaviors | QOLID |
| Umphonsathien [64] | 2022 | to analyse the impact of intermittent very-low calorie diet on glycemic, metabolic and QOL outcomes in people with T2DM | Thailand | UMIC | Structured dietary | 1. Calorie restricted diet (600 kcal per day) for two days/week 2. Calorie restricted diet (600 kcal per day) for four days/week, and ad libitum food consumption on non-restricted days in both groups | Usual standard diabetes care and normal diet of 1500 to 2000 Kcal/day | 124 | 14/14/12 | 0/0 | I 1: 49.5 (7.2) I 2: 47.6 (7.9) C: 52.0 (6.0) | 14/50/17 | QOL, HbA1c, FPG, BMI | SF-36 |
| Wattana [58] | 2007 | to assess the effects of a diabetes self-management education program on HbA1c, coronary heart disease risk and QOL among people with T2DM | Thailand | UMIC | SME | Education based on the theories of self-efficacy and self-management, followed by home visits | Usual nursing care that consisted of physical examination and general health education based on the institutional guideline | 26 | 79/78 | 5.0/7.6 | I: 58.4 (10.1) C: 55.1 (10.2) | 20/28 | HbA1c, QOL | SF-36 |
| Wichit [9] | 2017 | to determine the effects of a family-oriented education program in people with T2DM | Thailand | UMIC | SME | Family-oriented self-management intervention based on the self-efficacy theory | Usual care from clinical staff | 13 | 70/70 | 4.3/4.3 | I: 61.3 (11.6) C: 55.5 (10.5) | 24/30 | HbA1c, QOL, diabetes self-management | SF-12 |
| Wongrochananan [69] | 2015 | to study the effect of an interactive multi-modality intervention comprising of sms, email and website in enhancing self-management behavior in people with T2DM | Thailand | UMIC | SME | Use of interactive multi-modality (IMM) intervention consisting of website, email, and SMS to help improve self-management behaviors | Usual general education/ advice through email | 26 | 78/48 | 29.4/37.5 | I: 53.6 (8.6) C: 51.3 (7.9) | 56/50 | QOL, self-care behavior | DQOL |

(*Continued*)

**Table 1.** (Continued)

| Author | Year | Objective of the study | Study country | Country classification | Intervention type | Intervention detail | Control detail | Study duration (weeks) | Sample size (I/C) | Study drop-out (I/C) % | Age Mean (SD) | Gender (I % Male/C % Male) | Outcome measures of interest | QOL tool |
|---|---|---|---|---|---|---|---|---|---|---|---|---|---|---|
| Yang [74] | 2022 | to examine the effect of a telemedicine education approach in improving metabolic and QOL outcomes in people with T2DM | China | UMIC | SME | A WeChat public telemedicine management platform called "The home of Xinqiao nutrition" to monitor self-management behaviors and monthly education via telephone | Usual standard of care at a hospital | 65 | 50/50 | 6.0/0 | I: 65.1 (6.1) C: 67.3 (5.3) | 38/42 | QOL, HbA1c | SF-36 |
| Yucel [63] | 2015 | to determine the effects of pilates-based mat exercise on glycemic and psychological outcomes in people with T2DM | Turkey | UMIC | Structured exercise | Pilates based mat exercise involving warm-up; stretching; basic aerobic Pilates training for arms, legs, and body; and cool-down. | Usual routine medical care | 12 | 28/28 | 14.3/25.0 | I: 58.5 (7.0) C: 53.3 (9.0) | 0/0 | QOL, HbA1c, FBG | SF-36 |
| Zuo [59] | 2020 | to assess the effect of CBT on sleep disturbances and QOL in people with T2DM | China | UMIC | SME/CBT | Cognitive behavioral therapy with aerobic exercise | Usual care, i.e., face to face visit with the medical staff in a healthcare service | 39 | 96/95 | 2.0/2.1 | I: 63.9 (10.2) C: 61.7 (10.4) | 34/31 | QOL, HbA1c | Diabetes specific QOL scale |

*: Total sample size

#: Overall study drop-out rate.

UMIC: Upper-middle income country, LMIC: Lower-middle income country, SME: Self-management education, I/C: Intervention/Control, QOL: Quality of life, HbA1c: glycated haemoglobin, BMI: Body mass index, FBG: Fasting blood glucose, FPG: Fasting plasma glucose, SF: Short form, WBQ: Well-being questionnaire, WHOQOL-Bref: Abbreviated World Health Organization Quality of Life questionnaire, EQ5D: European quality of life-5D, QOLID: Quality of life instrument for Indian diabetes patients, DQOL: Diabetes Quality of Life, DCQOL: Diabetic Clients Quality of life, DQOL-BCI: Diabetes QOL Brief Clinical Inventory, SWED-QUAL: Swedish Health-Related Quality of Life Survey, ADDQOL: Audit of diabetes dependant Quality of Life, PAID: Problem Areas in Diabetes.

below, under each meta-analysis heading–"Meta-analysis of Mean QOL scores", "Meta-analysis of Physical Component Summary Scores", and "Meta-analysis of Mental Component Summary Scores".

### Effect of behavior interventions on quality of life

**Meta-analysis of Mean QOL scores.** Twenty-five studies reported overall QOL scores (participants n = 3,867). The health behavior interventions achieved a significant improvement in the QOL score compared to the control, with a large effect size (SMD: 1.62, 95% CI: 0.65 to 2.60 $I^2$: 0.96, p-value: 0.001) (Fig 2). Meta-analysis using different correlation coefficient (r = 0.4 and 0.8) did not change the direction of the effect or the heterogeneity (S6 Table).

The overall QOL score was not significantly different based on the subgroup analysis of intervention types (Q = 0.17, p-value = 0.63) or setting (Q = 0.49, p-value = 0.78) (S6 Table). However, interventions based in hospital/clinic (n = 21) and community health care (n = 4) significantly improved QOL compared to those based in home/web-based (although low number of studies included in this group, n = 2). Meta-regression analysis did not suggest a linear relationship between duration of intervention and mean QOL scores (Q = 1.13, p-value = 0.28). However, subgroup analysis suggested that interventions that were 12 weeks or longer

| Study | Total | Intervention Mean | SD | Total | Control Mean | SD | SMD | 95%-CI | Weight |
|---|---|---|---|---|---|---|---|---|---|
| Cheng (2019) | 121 | 0.26 | 1.53 | 121 | 0.19 | 1.58 | 0.04 | [-0.21; 0.30] | 3.7% |
| Butt (2015) | 37 | 7.35 | 13.28 | 36 | 6.52 | 14.66 | 0.06 | [-0.40; 0.52] | 3.7% |
| Azami (2018) | 71 | 0.34 | 8.15 | 71 | -0.35 | 7.93 | 0.09 | [-0.24; 0.41] | 3.7% |
| Abraham (2020) | 40 | 0.05 | 0.27 | 40 | 0.02 | 0.27 | 0.11 | [-0.33; 0.55] | 3.7% |
| Anderson (2009) | 156 | 9.10 | 17.33 | 154 | 3.70 | 18.42 | 0.30 | [0.08; 0.53] | 3.8% |
| Nazir (2020) | 196 | 0.13 | 0.28 | 196 | 0.01 | 0.29 | 0.42 | [0.22; 0.62] | 3.8% |
| Wongrochananan (2015) | 78 | 5.32 | 7.54 | 48 | 1.79 | 6.48 | 0.49 | [0.13; 0.86] | 3.7% |
| Saghaee (2020) | 17 | 4.41 | 6.91 | 17 | 0.82 | 7.36 | 0.49 | [-0.19; 1.17] | 3.7% |
| Akinci (2018)b | 22 | 0.16 | 0.16 | 22 | 0.06 | 0.22 | 0.51 | [-0.10; 1.11] | 3.7% |
| Jaipakdee (2015) | 203 | 5.70 | 5.38 | 200 | 2.30 | 5.84 | 0.60 | [0.41; 0.80] | 3.8% |
| Wattana (2007) | 79 | 9.82 | 13.41 | 78 | -0.67 | 13.40 | 0.78 | [0.45; 1.10] | 3.7% |
| Akinci (2018)a | 22 | 0.26 | 0.27 | 22 | 0.06 | 0.22 | 0.80 | [0.19; 1.42] | 3.7% |
| Cani (2015) | 37 | 5.15 | 12.73 | 41 | -3.83 | 8.79 | 0.82 | [0.36; 1.28] | 3.7% |
| Torabizadeh (2018) | 50 | 20.50 | 15.48 | 50 | 7.55 | 13.54 | 0.88 | [0.47; 1.29] | 3.7% |
| Rondhianto (2018) | 60 | 26.30 | 13.70 | 60 | 12.23 | 13.21 | 1.04 | [0.66; 1.42] | 3.7% |
| Mohammadi (2018) | 120 | 3.60 | 3.12 | 120 | 0.40 | 2.95 | 1.05 | [0.78; 1.32] | 3.7% |
| Rasoul (2019) | 49 | 21.30 | 2.88 | 49 | 18.10 | 3.15 | 1.05 | [0.63; 1.47] | 3.7% |
| Lyu (2021) | 54 | 106.33 | 63.25 | 52 | 37.01 | 59.80 | 1.12 | [0.71; 1.53] | 3.7% |
| Zuo (2020) | 96 | 12.30 | 11.31 | 95 | -0.40 | 10.30 | 1.17 | [0.86; 1.48] | 3.7% |
| Umphonsathien (2022)b | 14 | 616.00 | 123.87 | 12 | 121.00 | 551.95 | 1.25 | [0.39; 2.10] | 3.7% |
| Umphonsathien (2022)a | 14 | 313.00 | 123.87 | 12 | 121.00 | 133.63 | 1.45 | [0.57; 2.33] | 3.6% |
| Chaveepojnkamjorn (2009) | 80 | 15.60 | 6.14 | 84 | -0.70 | 8.65 | 2.15 | [1.77; 2.54] | 3.7% |
| Rias (2020)b | 20 | 14.76 | 6.18 | 21 | -0.35 | 6.67 | 2.30 | [1.50; 3.11] | 3.7% |
| Singh (2020) | 112 | 23.80 | 6.00 | 115 | 3.10 | 8.30 | 2.84 | [2.47; 3.21] | 3.7% |
| Peimani (2018) | 100 | 48.41 | 21.69 | 100 | -15.90 | 16.57 | 3.32 | [2.89; 3.75] | 3.7% |
| Shahsavari (2021) | 40 | 14.20 | 2.31 | 40 | -0.12 | 2.26 | 6.20 | [5.12; 7.28] | 3.6% |
| Safavi (2011) | 61 | 48.10 | 4.33 | 62 | -32.60 | 6.73 | 14.16 | [12.32; 15.99] | 3.3% |
| **Overall effect** | **1949** | | | **1918** | | | **1.63** | **[0.65; 2.60]** | **100.0%** |
| **Prediction interval** | | | | | | | | **[-3.75; 7.00]** | |

Heterogeneity: $I^2$ = 96% [96%; 97%], $\tau^2$ = 6.5604, $p$ < 0.01

-15 -10 -5 0 5 10 15
Favors Control Favors Intervention

**Fig 2. Forest plot comparing mean QOL scores of intervention and control group.**

significantly improved the overall QOL compared to interventions less than 12 weeks long (without a significant sub-group difference Q = 0.84, p-value = 0.36). Subgroup analyses based on types of QOL scales did not suggest significant differences in QOL between studies that used generic scale compared to those that used diabetes-specific scale (Q = 0.80, p-value = 0.37). Subgroup analysis based on methodology quality of included studies resulted in a significant improvement in mean QOL in interventions that deemed to have a 'good' quality of methodology (n = 21). Interventions with 'fair' method quality (n = 6) did not result in a significant improvement in mean QOL (n = 6). However, the test for subgroup analysis was not significant (Q = 0.66, p-value = 0.41) (S6 Table).

In the GRADE certainty of evidence assessment, the studies were downgraded one level each due to high heterogeneity ($I^2$: 94%) and the presence of publication bias as seen in the funnel plot (S2 Fig). However, due to large effect size (SMD: 1.62), the studies were upgraded by one level. Overall, the GRADE certainty of evidence was classified as moderate quality (S5 Table).

Funnel plot (S2 Fig) showed degree of asymmetry, aligned with the high heterogeneity observed.

**Meta-analysis of Physical Component Summary scores.** Seven studies reported PCS scores (participants n = 834). The effect of health behavior interventions on PCS score did not reach a statistical significance level compared to the control group (SMD: 0.76, 95% CI: -0.03 to 1.56 $I^2$: 0.94, p-value: 0.060) (Fig 3).

Overall PCS score was not significantly different based on the subgroup analysis of intervention types (Q = 0.05, p-value = 0.82). Subgroup analysis based on settings resulted in a significant difference (Q = 37.52, p-value<0.001), with the subgroup of community health centre resulting in a large significant effect size (SMD: 3.22, 95% CI: 2.26, 4.17), but this was based on only one study. Overall PCS score was not significantly different based on the subgroup analysis of intervention duration (Q = 0.29, p-value = 0.58) (S6 Table). Subgroup analysis of overall PCS scores based on methodology quality of included studies was not different between studies that deemed to have a 'good' methodology quality (n = 7) compared to the one study that had a fair methodology quality (S6 Table). Meta-regression analysis also did not suggest a linear relationship between duration of the behavior change intervention and changes in the PCS scores (Q = 0.36, p-value = 0.546).

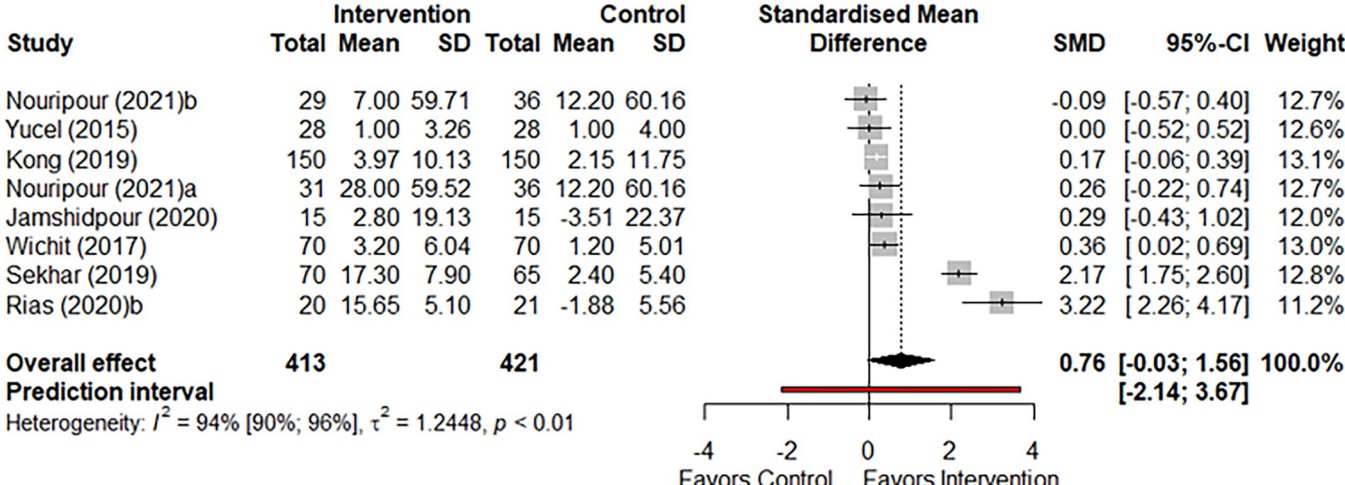

**Fig 3. Forest plot comparing mean PCS score of intervention and control group.**

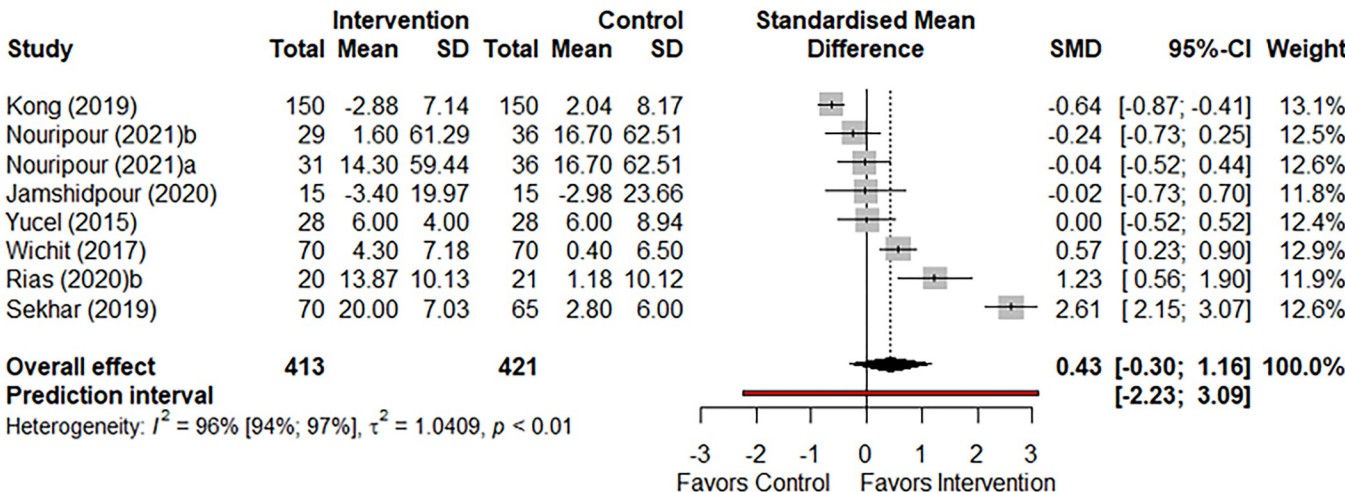

**Fig 4. Forest plot comparing mean MCS score of intervention and control group.**

Due to high heterogeneity ($I^2$: 94%) and imprecise results (wide CI), the GRADE certainty of evidence was classified as low quality (S5 Table).

Funnel plot (S3 Fig) also showed some degree of asymmetry, aligned with the high heterogeneity observed.

**Meta-analysis of Mental Component Summary scores.** Seven studies reported MCS scores (participants n = 834). The effect of health behavior interventions on the MCS score did not reach statistical significance compared to the control group (SMD: 0.43, 95% CI: -0.30 to 1.16 $I^2$: 0.94, p-value: 0.249) (Fig 4).

Overall, the MCS score was not significantly different based on the subgroup analysis of intervention types (Q = 0.40, p-value = 0.48). Subgroup analysis based on settings resulted in a significant difference (Q = 30.60, p-value<0.001), with subgroup of community health centre resulting in a large significant effect size (SMD: 1.23, 95% CI: 0.56, 1.90), but this was based on only one study. Overall, the MCS score was not significantly different based on the subgroup analysis of intervention duration (Q = 0.05, p-value = 0.814) (S6 Table). Subgroup analysis of overall MCS scores based on methodology quality of included studies was not different between studies that deemed to have a 'good' methodology quality (n = 7) compared to the one study that had a fair methodology quality (S6 Table). Meta-regression analysis also did not suggest a linear relationship between duration of the behavior change intervention and changes in the MCS scores (Q = 0.035, p-value = 0.350).

Due to high heterogeneity ($I^2$: 94%) and imprecise results (wide CI), the GRADE certainty of evidence was classified as low quality (S5 Table).

Funnel plot (S4 Fig) of studies reporting MCS scores showed some degree of asymmetry, aligned with the high heterogeneity observed.

Overall, fourteen studies were excluded from the meta-analysis because they reported domain specific QOL scores, but not the total mean QOL scores or composite (PCS or MCS) scores.

## Effect of health behavior interventions on secondary outcomes

Since the primary aim of this study was to systematically review and meta-analyse QOL outcomes, secondary outcomes were not meta-analysed. However, we did find that 25 out of 45 studies included in the review reported reduction in HbA1c in the intervention group. The

reduction in 13 (52%) of those studies was statistically significant. Seventeen studies reported reduction in BMI in the intervention group, six of which were statistically significant. Similarly, seven studies reported changes in FBG, with three demonstrating a significant reduction in the intervention group. Four studies reported non-significant improvement in dietary behavior and six studies reported improvement in physical activity behavior, where one showed a statistically significant improvement. Two studies showed significant improvement in medication adherence in the intervention group. Three studies showed improvement in foot care in the intervention group, although not statistically significant.

## Discussion

This systematic review analysed and synthesized the evidence from 45 RCTs to assess the impact of health behavior interventions to manage T2DM on the QOL of people in LMICs. Overall, the review demonstrated great heterogeneity in terms of type of intervention, description and duration of intervention, and the outcome measures. Nevertheless, the review suggested that health behavior interventions for improving T2DM management can improve QOL, along with glycemic outcome and self-care behaviors.

Our meta-analysis found a significant improvement in the overall QOL scores with a large effect size of 1.62 (95% CI: 0.65 to 2.60). This is comparable to previous meta-analyses of DSME interventions that showed statistically significant improvement in the QOL, albeit with small effect sizes of 0.26 [78] and 0.28 [79] respectively. Improved health behaviors can prevent diabetes comorbidity and complications as patients need to take less medications and worry less about the care associated with the complications [80]. Furthermore, positive health behaviors such as regular physical activity have shown to improve insulin sensitivity [81], which is significantly associated with QOL [82]. Another explanation to improved QOL could be improved self-management behaviors attributed to self-efficacy, as studies have shown that positive behavior change among people with diabetes is mediated by self-efficacy, which is found to have a strong positive correlation with QOL [83,84].

Our meta-analysis including seven studies didn't find any significant changes in the PCS (SMD: 0.76, 95% CI: -0.03 to 1.56) or MCS (SMD: 0.43, 95% CI: -0.30 to 1.16) measures of QOL. The limited number of studies presenting the PCS and MCS scores, small sample size and short intervention duration of the trials might explain these findings. The eight subscales of the SF questionnaire—physical functioning, role physical, bodily pain, general health, vitality, social functioning, role emotional, and mental health contribute in different proportions in the calculation of the PCS and MCS scores, which involves a special scoring algorithm. However, there have been concerns regarding the validity of the scoring method used to generate PCS and MCS scores [85,86]. Furthermore, the developers of SF questionnaire do not approve of combining these summary measures to generate a total QOL score [87], hence, many researchers tend to present only the eight-subscale data and not calculate the physical and mental summary measures [37,56,66,74].

Sub-group analysis based on intervention setting showed a significant difference in the PCS (d = 3.22, Q = 37.52, p-value<0.001) and MCS (d = 1.22, Q = 30.60, p-value<0.001) scores for the subgroup of studies conducted in community health centres. This could imply that an intervention delivered at community health centres is more effective in improving the QOL in the physical and mental dimensions of health. However, these analyses had only one study each, hence, it warrants caution in interpretation. The finding is congruent to a review which suggested that health education intervention delivered at a community health setting is more effective in improving QOL and self-management behaviors compared to patient-focused hospital setting [88].

In relation to secondary outcomes, our findings indicated that health behavior interventions are effective in improving T2DM patients' glycemic, anthropometric and self-care measures. Statistically significant reductions were seen in metabolic outcomes (HbA1c, FBG), also supported by multiple reviews [6,15,89]. Similarly, dietary behavior and physical activity were found to have improved post-interventions, but not significantly. Studies have questioned whether diet change plans are sustainable and implementable in the long run, which may explain why no significant improvement was seen in dietary behavior [90,91]. In addition, the low number of studies that reported dietary and physical activity outcomes in this review might explain the lack of statistical significance. Statistically significant improvement in medication adherence, as seen in this review, could be attributed to the combined educational/behavioral intervention strategy designed and implemented to improve patient knowledge and awareness on medication adherence for better diabetes outcomes [92].

## Strengths and limitations

Our study has several strengths. It is the first systematic review and meta-analysis to study and synthesize available evidence on the effect of health behavior interventions on QOL in people with T2DM in LMICs. A relatively large sample size (n = 8336) in this review ensures greater reliability that behavioral interventions are indeed effective in improving the QOL in people with T2DM. This also addresses the limitation of previous meta-analyses only having a secondary focus on QOL outcomes, whose interpretation of change in QOL was limited due to a much smaller sample size (n = 2645) [78]. Another strength is that all the studies included in this review and meta-analysis were RCTs, as this strengthens the internal validity and minimises the chances of confounders [93]. Similarly, not using LMIC search terms in our search strategy, but rather searching for the studies conducted worldwide and later differentiating the retrieved studies based on income regions may have affected the number of studies we retrieved. We believe this search strategy generated more results, hence allowing us to more thoroughly examine studies conducted worldwide and extract the ones that met the objective of this study in terms of setting.

This study also has limitations. Firstly, as expected, the heterogeneity of the studies was high as the scales used to measure QOL in the RCTs varied widely. The complexity of QOL constructs and lack of a standard definition has resulted in a wide variety of tools being used in the QOL measurement. Secondly, we excluded the studies that only presented the domain-specific scores instead of PCS, MCS or overall scores in the meta-analysis. Thirdly, the findings of this meta-analysis only related to short-term QOL of participants (i.e., immediate post-intervention effects) and any lasting/sustained effect was not examined in this review as there was no sufficient data to examine longer term effects on QOL. Similarly, the potential of reporting bias in the included RCTs may have influenced the findings of our review. Only including studies published in English language in the review is another limitation. Lastly, the limited number of interventions measuring QOL outcomes hinders the possibility of comparing our findings with prior studies. Therefore, more RCTs should include QOL measures to enable future researchers to pool studies that have used similar QOL measurement approaches in their meta-analyses. Furthermore, large and well-powered RCTs of high methodological quality are necessary to establish the effect of health behavior interventions on QOL.

## Conclusion

In conclusion, this systematic review and meta-analysis identified health behavior interventions targeted at improving T2DM management as effective strategies in improving the QOL of people with T2DM. The study demonstrated a moderate certainty of evidence for the overall QOL findings, indicating that the immediate post-intervention improvement on the overall QOL of people with

T2DM is likely to be close to the true effect of the interventions on QOL over the short term. However, the low quality of evidence of PCS and MCS findings limits the strength of our conclusion from the pooled evidence. Due to the variability of the scales used in QOL measurement, the interpretation of our findings warrants caution. Studies with specific standardized scales are recommended for more robust estimates. Further research is needed to examine the long-term effectiveness of health behavior interventions on QOL. Furthermore, this review highlights the need for more well-designed RCTs of high methodological quality focusing on QOL as a primary outcome measure. The evidence generated from this review may guide future research and clinical practice and may derive policy implications for the management of T2DM in LMICs.

## Supporting information

**S1 Checklist. PRISMA checklist.**
(DOC)

**S1 Fig. R codes used in the meta-analysis.**
(DOCX)

**S2 Fig. Funnel plot of Mean QOL.**
(DOCX)

**S3 Fig. Funnel plot of Mean Physical Component Summary.**
(DOCX)

**S4 Fig. Funnel plot of Mean Mental Component Summary.**
(DOCX)

**S1 Table. Detailed search strategy.**
(DOCX)

**S2 Table. Intervention characteristics of studies included in the review sorted alphabetically by author.**
(DOCX)

**S3 Table. Effect of health behavior intervention on the primary (quality of life) and secondary outcomes.**
(DOCX)

**S4 Table. Quality assessment of studies using Joanna Briggs Institute (JBI) Critical Appraisal Checklist for Randomized Controlled Trials.**
(DOCX)

**S5 Table. GRADE certainty of evidence.**
(DOCX)

**S6 Table. Subgroup analyses.**
(DOCX)

**S7 Table. List of studies excluded during full-text screening.**
(DOCX)

## Acknowledgments

The authors would like to acknowledge the research librarian of Central Queensland University for providing support in building search strategy.

## Author Contributions

**Conceptualization:** Ashmita Karki.

**Data curation:** Ashmita Karki.

**Formal analysis:** Ashmita Karki, Saman Khalesi.

**Methodology:** Ashmita Karki, Padam Dahal.

**Supervision:** Corneel Vandelanotte, Lal B. Rawal.

**Writing – original draft:** Ashmita Karki.

**Writing – review & editing:** Ashmita Karki, Corneel Vandelanotte, Saman Khalesi, Padam Dahal, Lal B. Rawal.

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
