## [Decision Letter · Decision Letter 0]

2 Jun 2023

PONE-D-23-05175The effect of health behavior interventions to manage Type 2 diabetes on the quality of life in low-and middle-income countries: a systematic review and meta-analysisPLOS ONE

Dear Dr. Karki,

Thank you for submitting your manuscript to PLOS ONE. After careful consideration, we feel that it has merit but does not fully meet PLOS ONE’s publication criteria as it currently stands. Therefore, we invite you to submit a revised version of the manuscript that addresses the points raised during the review process.

 Please submit your revised manuscript by Jul 17 2023 11:59PM. If you will need more time than this to complete your revisions, please reply to this message or contact the journal office at plosone@plos.org. Please include the following items when submitting your revised manuscript:A rebuttal letter that responds to each point raised by the academic editor and reviewer(s). You should upload this letter as a separate file labeled 'Response to Reviewers'.A marked-up copy of your manuscript that highlights changes made to the original version. You should upload this as a separate file labeled 'Revised Manuscript with Track Changes'.An unmarked version of your revised paper without tracked changes. You should upload this as a separate file labeled 'Manuscript'.

We look forward to receiving your revised manuscript.

Kind regards,

Edward Zimbudzi

Academic Editor

PLOS ONE

Journal Requirements:

Reviewers' comments:

Reviewer's Responses to Questions

**Comments to the Author**

1. Is the manuscript technically sound, and do the data support the conclusions?

Reviewer #1: Partly

Reviewer #2: Yes

2. Has the statistical analysis been performed appropriately and rigorously? 

Reviewer #1: No

Reviewer #2: Yes

3. Have the authors made all data underlying the findings in their manuscript fully available?

Reviewer #1: Yes

Reviewer #2: Yes

4. Is the manuscript presented in an intelligible fashion and written in standard English?

Reviewer #1: Yes

Reviewer #2: Yes

5. Review Comments to the Author

Reviewer #1: Thank you for asking me to perform this review. I hope that my comments and suggestions can help the authors improve the reporting of their review, and strengthen the synthesis and conclusions/interpretations.

1. Background: The statement using reference 5 seems inflated without any mention of its context eg high income countries or such. Findings from what appears to be a similar review to this (Ref 13) seem to be crucial for providing rationale for this review; perhaps describe how/why findings were inconclusive and/or any other limitations of that review or the eligible studies. The reviews from the Middle East and West Pacific should be described as only including studies from the respective countries (if this is true) to enhance the rational for this current (broader scope) review. Line 79, I think “reviews” not “studies” are limited?

2. The primary outcome is described (in several places) as the “change in QOL scores between the intervention and control arms”, but should be described as the difference between arms in the change scores. I would encourage the authors to run a post hoc subgroup analysis comparing findings between generic and disease-specific overall QOL scales because these can capture quite different domains/concepts and disease-specific scores may be more sensitive to intervention effects. Further, if effects are similar between groups this may reduce the limitations mentioned for use of so many measurement tools.

3. Were only studies that reported QOL outcomes included, for example if a study protocol stated that they measured the outcome but there is no report of the findings were they excluded? If this is the case for any studies I would encourage the authors to include those studies and account for this potential reporting bias/missing data. It is somewhat concerning that only studies having pre and post measurement of QOL (ie for change scores) were included and the limitations of this should be noted. When reviewing RCTs it is usually thought that change scores are not required/superior since baseline imbalances should be minimal. Some rationale for this criteria should be mentioned.

4. Please include the date of the searches and whether there were any date limits for eligibility (in abstract and methods section). If at all feasible within word limits, in the abstract mention that data extraction was single reviewer with verification and that risk of bias (preferable term to quality assessment) assessment was conducted in duplicate. In then the results section of the main text, please mention and (in an appendix) include a list of excluded full texts; this can be limited to the ones potentially most relevant to the topic if necessary. Tables 2 and 3 could be moved to an appendix, with the addition to Table 1 of the QOL scale(s) used by each study; Table 1 could be sorted by one or more logical factors, such as alphabetic names, study date, types of intervention etc., with an indication of this method in the text or table header/title. In Table 3, I’m not sure the data from subscales is useful since it wasn’t synthesized in any manner in the review.

5. What did the authors do if QOL was measured at more than one time point post-intervention? Do the author think that post-intervention follow-up duration is variable enough (as was intervention duration) do run any analysis on this?

6. For quality assessment/risk of bias, i) allocation to study groups occurs before the intervention is delivered can always be concealed, ii) the (in)feasibility of blinding participants to their group (after allocation) does not mean that there is no risk for bias – the ratings should be lowered for lack of participant blinding, especially for a review of patient-reported outcomes. The lack of blinding may be lessened to some degree in studies with wait-list or attention controls, if these existed. What does appropriate trial design mean? Also, if the review authors essentially re-analyzed the study authors data (i.e. not relying on the trial authors adjusted analysis etc) they should not use (or heavily weigh) the authors’ analysis as criteria for their own assessment. The assessments should be used in the review synthesis to some extent (see below on possibly sensitivity analysis). The results section has very little description apart from loss to follow-up related to the risk of bias across studies, and does not mention where to find the full assessments.

7. It is assumed that the control groups were all usual care/no intervention, but this should be stated in the study eligibility. In some cases the usual care group (in our experience with chronic diseases) can closely align with one of the interventions of interest to the review. The authors should describe the usual care arms, as able, across the studies and consider if there needs to be any sensitivity analysis or study exclusions based on these.

8. Analysis: Current methods guidance for systematic reviews supports sensitivity analysis for methodological decisions of the review authors (e.g. use of one vs another type of analysis, need to impute variance measures from other studies in the review when they are missing from one or more studies) or concerns about study reporting or eligibility (e.g. uncertain eligibility of an intervention, high risk of bias), rather than based on the study results as is done in this review for their influencer and outlier analysis – for which the possible cause of any changes to the effect estimate are not interpretable. I would encourage the authors to consider removing their influencer and outlier analyses and focus on review or study-level variables that may explain the heterogeneity. They could add sensitivity analysis for risk of bias (removing high ROB studies) and possibly from use of a different correlation coefficient for their calculations of SDs for change scores (as Cochrane handbook recommends). The methods section does not describe what covariates (including whether they were categorical or continuous) were used for meta-regressions; it also does not mention that intervention duration (< vs > 12 weeks) was used for subgroup analysis.

9. Please describe in the results section why there were 14 studies excluded from the meta-analysis, presumably because only subscales of QOL tools were reported? If there were other reasons such as missing variance measures, these can be dealt with by imputing values from other studies and running sensitivity analysis.

10. Interpretation: The findings from the subgroup analysis of delivery setting are heavily overstated in the abstract and main text conclusions. In the abstract conclusions, the authors should likely state that “may substantially improve” or similar, to impart the effect size, rather than just mentioning “significance”. It is unfortunate that the authors did not assess certainty of the evidence (using GRADE or similar) which may have changed some of their conclusions (e.g. “borderline” significant findings for the PCS may have been interpreted differently i.e. possibly a moderate effect size with limitations/reduced certainty due to imprecision and/or inconsistency [causing the wide CI]; risk of bias more integrated into their interpretations if indeed this made any difference to findings). If this is not possible, adding a couple sensitivity (e.g. risk of bias) and subgroup (eg type of QOL scale) may allow for more comment on the robustness of the results and discussion on the specific limitations. Systematic reviews usually use duplicate full text screening and the lack of this should be mentioned in the limitations section.

Reviewer #2: Thank you authors for doing a great job on the manuscript. Kindly respond to comments made especially on the methodology section. Again, specify which systematic review methodology you followed. Is it according to JBI format or ?

6. PLOS authors have the option to publish the peer review history of their article (what does this mean?). If published, this will include your full peer review and any attached files.

Reviewer #1: No

Reviewer #2: No

---

## [Author Response · Author response to Decision Letter 0]

17 Jul 2023

Editor’s comments:

We thank the editors for this comment. We have revised our manuscript aligning with the style requirements of PLOS One. We have named our files aligning with the file naming requirement of PLOS One.

We thank the editors for this comment. The “data not provided” phrase in the manuscript has been removed. (please see page 28, line 237)

We thank the editors for this comment. We have now moved the ethics statement to the Method section of the manuscript. Please see page 6 lines 100-101. The statement reads as follows:

“Ethical approval was not required for conducting this systematic review and meta-analysis.”

We thank the editors for this comment. We have now included the captions for our supporting information files at the end of our manuscript under the heading “Supporting information” (please see page 45 lines 770-789) and updated the in-text citations to match accordingly. 

Reviewers’ comments

Reviewer #1: Thank you for asking me to perform this review. I hope that my comments and suggestions can help the authors improve the reporting of their review, and strengthen the synthesis and conclusions/interpretations.

1. Background: The statement using reference 5 seems inflated without any mention of its context eg high income countries or such. Findings from what appears to be a similar review to this (Ref 13) seem to be crucial for providing rationale for this review; perhaps describe how/why findings were inconclusive and/or any other limitations of that review or the eligible studies. The reviews from the Middle East and West Pacific should be described as only including studies from the respective countries (if this is true) to enhance the rational for this current (broader scope) review. Line 79, I think “reviews” not “studies” are limited?

Thank you so much for your thoughtful comments. As advised, we have revised the background section, and added the context of the finding (please see page 4, lines 57-59). The revised statement now reads as follows:

“Furthermore, improved QOL is significantly associated with improved diabetes self-management as shown by a review of studies mostly conducted in high income countries (HICs).”

Similarly, we have revised the background section describing why findings in relation to QOL were inconclusive in the study by Flood et. al (ref 13) (please see page 4, lines 61-65). The revised paragraph now reads as follows:

“Previous reviews from LMICs have shown the effectiveness of DSME on metabolic parameters [6, 12], however, outcomes in relation to QOL are inconclusive due to limited data availability on QOL outcomes [13]. This paucity of data due to the complex construct of QOL combined with likely measurement error makes the assessment of QOL really challenging [14].”

The reviews from the Middle East and West Pacific have been described as reviews of studies conducted in either LMICs or HICs. Similarly, we have also stated the limitations of previous reviews in the revised manuscript to enhance the rational of our review (please see page 4-5, lines 65-75). The revised paragraph now reads as follows:

“A review of twelve studies, conducted in four LMICs and two HICs indicated that DSME interventions were effective in improving clinical parameters as well as patient-reported outcomes such as self-management behavior, QOL and self-efficacy [15]. Similarly, another review of eight studies, conducted in five HICs and three LMICs demonstrated QOL improvement in 3 out of the 5 randomized controlled trials (RCTs) that reported change in QOL measures [16]. However, these reviews only included studies with DSME components, and excluded studies solely focussing on other health behaviour interventions such as structured dietary and/or exercise programs. In addition, these reviews have not been synthesized to produce a pooled estimate of the effect size, which is critical to strengthen the existing evidence base. Moreover, there is a dearth of reviews exclusively focusing on LMICs.”

The sentence on line 73 (originally line 79) has been revised to suggest “reviews”, not “studies” (please see page 5, line 73). The sentence now reads as follows:

“In addition, these reviews have not been synthesized to produce a pooled estimate of the effect size, which is critical to strengthen the existing evidence base. Moreover, there is a dearth of reviews exclusively focusing on LMICs.

2. The primary outcome is described (in several places) as the “change in QOL scores between the intervention and control arms”, but should be described as the difference between arms in the change scores. I would encourage the authors to run a post hoc subgroup analysis comparing findings between generic and disease-specific overall QOL scales because these can capture quite different domains/concepts and disease-specific scores may be more sensitive to intervention effects. Further, if effects are similar between groups this may reduce the limitations mentioned for use of so many measurement tools.

Thank you so much for the comment. We have now replaced “Change in QOL” with “difference in QOL change scores between the intervention and control groups” as the primary outcome of the review throughout the manuscript. The revised sentences now read as follows:

“Difference in QOL change scores between the intervention and control group was calculated as the standardized mean difference (SMD) of QOL scores observed between the intervention and control groups.” (page 2 line 27-29)

“The primary outcome measure of this review was difference in QOL change scores between the intervention and control groups.” (page 8 line 147-148)

We conducted subgroup analyses based on QOL scales and have added to the methods section (please see page 10, line 201-205). The statement reads as follows:

“Subgroup analyses were performed to investigate differences in the effect of behavior change intervention on QOL scores based on intervention type (DSME, diet and/or physical activity interventions), and settings (hospital/diabetes clinic, community health centere, home/web-based), intervention duration (less than 12 weeks, 12 weeks or longer), QOL scale (generic, disease-specific), and methodology quality (good, fair, poor).”

We have added the findings from sub-group analyses to the results section (please see page 31, line 302-304). The added findings read as follows:

“Subgroup analyses based on types of QOL scales did not suggest significant differences in QOL between studies that used generic scale compared to those that used diabetes-specific scale (Q=0.80, p-value=0.37).”

Subgroup analysis based on QOL scales for PCS and MCS was not possible because all studies involved used a generic QOL scale. We have added this to the supplementary table (S6 Table) footnote. 

3. Were only studies that reported QOL outcomes included, for example if a study protocol stated that they measured the outcome but there is no report of the findings were they excluded? If this is the case for any studies I would encourage the authors to include those studies and account for this potential reporting bias/missing data. It is somewhat concerning that only studies having pre and post measurement of QOL (ie for change scores) were included and the limitations of this should be noted. When reviewing RCTs it is usually thought that change scores are not required/superior since baseline imbalances should be minimal. Some rationale for this criteria should be mentioned.

Thank you for pointing this out. We did not see any protocols that stated they measured the outcome but didn’t report the findings. We acknowledge that there might be some degree of reporting bias in the literature as previous studies suggest that reporting bias does occur in QOL literature (1, 2). Therefore, we have now highlighted this as a limitation of the study (please see page 37 line 453-455). In this review, our objective was to ascertain the effect of the health behaviour intervention on QOL as the primary outcome, therefore, by design, we reviewed both the pre-and post-intervention QOL scores and the effect therein of the intervention on the study-assessed QOL outcomes (either post-intervention QOL scores compared against control or simply changes in QOL scores post-intervention). The revised limitation sub-section reads as follows:

“Similarly, the potential of reporting bias in the included RCTs may have influenced the findings of our review.”

4. Please include the date of the searches and whether there were any date limits for eligibility (in abstract and methods section). If at all feasible within word limits, in the abstract mention that data extraction was single reviewer with verification and that risk of bias (preferable term to quality assessment) assessment was conducted in duplicate. In then the results section of the main text, please mention and (in an appendix) include a list of excluded full texts; this can be limited to the ones potentially most relevant to the topic if necessary. Tables 2 and 3 could be moved to an appendix, with the addition to Table 1 of the QOL scale(s) used by each study; Table 1 could be sorted by one or more logical factors, such as alphabetic names, study date, types of intervention etc., with an indication of this method in the text or table header/title. In Table 3, I’m not sure the data from subscales is useful since it wasn’t synthesized in any manner in the review.

We have added the date of searches and date limits in the abstract (please see page 2, lines 24-25). The revised sentence in the abstract reads as follows:

“Electronic databases PUBMED/MEDLINE, SCOPUS, CINAHL, Embase, Web of Science and PsycINFO were searched from May to June 2022. Studies published between January 2000 and May 2022, conducted in LMICs using randomized controlled trial design, using a health behavior intervention for T2DM management, and reporting QOL outcomes were included.”

Due to the abstract word limit, we couldn’t add that data extraction was done by a single reviewer and risk of bias assessment by two reviewers. However, we have presented the detailed information on assessment of risk of bias in the methods section main text field. 

We have included the list of excluded full text articles with reasons in an appendix (S7 Table) and also mentioned it in the text (please see page 11 line 217-218). The added sentences read as follows:

“Altogether, 386 studies were excluded after full text review. The list of excluded articles is presented in the appendix as S7 Table.”

As per your suggestion, we have moved Table 2 and Table 3 to appendix, S2 Table and S3 Table respectively, and added a column on Table 1 presenting the QOL scales which was previously in Table 2 (please see Table 1, pages 12-28). 

We have also added a column “Control detail” for providing information on what the control groups received. Table 1 has been sorted by author name in an alphabetic manner and we have indicated it in the title of the table (please see title of Table 1 on page 12, line 236). 

We presented the data from QOL subscales in Table 3 although we did not synthesize the subscale data. This was done so that our readership can have an overall idea of all QOL-related data reported by all studies included in the review, not only the QOL scale data that we used in our synthesis. Many trials have presented only the domain-specific QOL data, not the composite scores (PCS and MCS) or the overall mean QOL scores, hence, to provide an overall picture of all QOL data that studies have reported, we decided to keep the data from subscales in the table (S3 Table after revision).

5. What did the authors do if QOL was measured at more than one time point post-intervention? Do the author think that post-intervention follow-up duration is variable enough (as was intervention duration) do run any analysis on this?

Thank you, we appreciate the reviewer’s comment. For studies that measured QOL at multiple time point post-intervention, we only included the post-intervention QOL measures (directly following the intervention completion). As this study aimed to investigate the effect of health behaviour intervention on QOL and to reduce the follow-up effects (lost to follow-up, changes in behaviour, etc.) and to improve the homogeneity of data extracted, no follow-up data was included in our meta-analysis. Only five studies (Azami (2018), Mohammadi (2018), Zuo (2020), Cheng (56), Abraham (2020)) reported follow up measurements with follow-up durations ranging from 13 weeks to 26 weeks post-intervention. The overall meta-analysis effect of mean QOL was not sensitive to these studies (SMD=1.89, 95% CI: 0.71 to 3.08 I2:96.8). No further analyses were feasible. This is added to Methods (please see page 10 lines 195-198):

“For studies with QOL measurements at multiple time-points, only the measurements at baseline and following intervention completion (post-intervention) were included in the meta-analysis to reduce the follow-up effect and improve homogeneity in data extraction.”

6. For quality assessment/risk of bias, i) allocation to study groups occurs before the intervention is delivered can always be concealed, ii) the (in)feasibility of blinding participants to their group (after allocation) does not mean that there is no risk for bias – the ratings should be lowered for lack of participant blinding, especially for a review of patient-reported outcomes. The lack of blinding may be lessened to some degree in studies with wait-list or attention controls, if these existed. What does appropriate trial design mean? Also, if the review authors essentially re-analyzed the study authors data (i.e. not relying on the trial authors adjusted analysis etc) they should not use (or heavily weigh) the authors’ analysis as criteria for their own assessment. The assessments should be used in the review synthesis to some extent (see below on possibly sensitivity analysis). The results section has very little description apart from loss to follow-up related to the risk of bias across studies, and does not mention where to find the full assessments.

Thank you so much for the comment.

i) Indeed, allocation concealment can always be done in RCTs. However, in 26 RCTs included in our review, there was either a clear statement of “allocation concealment not being done” or no clear indication whether allocation was concealed or not. Some of those RCTs mentioned about generating random numbers with the use of computer (allocation sequence generation) (e.g., Mash (2014), Umphonsathien (2022), Yang (2022), Shi (2018)), however, they did not mention about concealment of those allocation sequences, hence, the allocation concealment for those RCTs have been rated as unclear (please see S4 Table in the appendix). Three RCTs clearly stated that allocation concealment was not done in their full texts (Abraham (2020), Arovah (2018), and Saghaee (2020)). 

ii) Since the type of intervention used in the RCTs included in our review are health behavior interventions to manage Type 2 diabetes, blinding of participants or study personnel wouldn’t have been generally feasible in the RCTs due to the involvement of participants and/or personnel in the behavioral interventions (3), hence we did not downgrade for those risks/uncertainties (4). Appropriate trial design means the design of the RCTs included in the review were appropriate for the objective of the study. For example, if an RCT is aiming to assess the effectiveness of DSME intervention in improving QOL of people with T2DM, then the appropriate RCT design would be parallel-group RCT or cluster RCT, but not crossover RCT (5). All RCTs included in our review followed appropriate trial design.

We have updated the results section to include clearer information on risk of bias within studies and certainty of evidence across studies included in the meta-analyses. The revised paragraphs read as follows:

“An overview of the quality assessment of the studies is presented in the appendix as S4 Table. Allocation of study participants to intervention and control groups wasn’t concealed in three studies and not clearly indicated in 23 studies. Only two studies clearly specified blinding of participants, five stated blinding of those delivering the intervention, and 13 specified blinding of outcome assessors. Blinding of participants is generally difficult in behavioral interventions due to participants being aware whether they receive the intervention, hence studies that did not address or clarify blinding were not downgraded for those risks or uncertainties [28]. Use of appropriate trial design and statistical analysis methodology was strong in all studies. Thirty-eight studies provided information on participants’ attrition with reasons for dropouts and withdrawals. In 23 studies, the data were analysed following the intention-to-treat principle. The quality of evidence across studies included in the meta-analyses, assessed using the GRADE approach, is presented in the appendix as S5 Table, and the description is provided below, under each meta-analysis heading – “Meta-analysis of Mean QOL scores”, “Meta-analysis of Physical Component Summary Scores”, and “Meta-analysis of Mental Component Summary Scores”.” (please see page 30, lines 268-282)

“In the GRADE certainty of evidence assessment, the studies were downgraded one level each due to high heterogeneity (I2: 94%) and the presence of publication bias as seen in the funnel plot (S3 Fig). However, due to large effect size (SMD: 1.62), the studies were upgraded by one level. Overall, the GRADE certainty of evidence was classified as moderate quality (S5 Table).” (please see page 31 lines 309-312)

“Due to high heterogeneity (I2: 94%) and imprecise results (wide CI), the GRADE certainty of evidence was classified as low quality (S5 Table).” (please see page 32, lines 337-338)

“Due to high heterogeneity (I2: 94%) and imprecise results (wide CI), the GRADE certainty of evidence was classified as low quality (S5 Table).” (please see page 34 lines 362-363)

7. It is assumed that the control groups were all usual care/no intervention, but this should be stated in the study eligibility. In some cases the usual care group (in our experience with chronic diseases) can closely align with one of the interventions of interest to the review. The authors should describe the usual care arms, as able, across the studies and consider if there needs to be any sensitivity analysis or study exclusions based on these.

Thank you for the comment. The control groups were either usual care, waitlist control or attentional control. This has been stated in the main text (please see page 6 lines 106-109) and the sentence reads as follows:

“The population group was people with T2DM; intervention was any health behavioural intervention targeted at T2DM management; comparison group was usual care, waitlist or attentional control group; and study design was randomized controlled trial.”

The control groups in the studies included in the review received usual, waitlist or attentional care in relation to routine diabetes care/education/management but did not receive any component of the intervention of interest except for one study by Castillo-Hernandez et al. The influence analysis (based on one-out method) did not suggest that the overall meta-analysis results are sensitive to the study by Castillo-Hernandez et al.

We have described the usual care arm in Table 1 across the studies (please see “Control detail” column in Table 1, pages 12-28). 

8. Analysis: Current methods guidance for systematic reviews supports sensitivity analysis for methodological decisions of the review authors (e.g. use of one vs another type of analysis, need to impute variance measures from other studies in the review when they are missing from one or more studies) or concerns about study reporting or eligibility (e.g. uncertain eligibility of an intervention, high risk of bias), rather than based on the study results as is done in this review for their influencer and outlier analysis – for which the possible cause of any changes to the effect estimate are not interpretable. I would encourage the authors to consider removing their influencer and outlier analyses and focus on review or study-level variables that may explain the heterogeneity. They could add sensitivity analysis for risk of bias (removing high ROB studies) and possibly from use of a different correlation coefficient for their calculations of SDs for change scores (as Cochrane handbook recommends). The methods section does not describe what covariates (including whether they were categorical or continuous) were used for meta-regressions; it also does not mention that intervention duration (< vs > 12 weeks) was used for subgroup analysis.

Thank you for the comment. Following previous literature (4), quality of the studies was rated as good (≥8), fair (6-7), or poor (≤5) based on the summary scores obtained from the critical appraisal using the JBI checklist for RCTs. Altogether 34 studies were of good quality and 11 studies of fair quality. Since no studies were of poor quality, we included all 45 studies in the review. Please see S4 Table in the appendix for the full quality assessment of the studies. 

Initially, in the manuscript, we had not presented the quality (good, fair, poor) data or grading of bias in the quality assessment table because JBI checklist for RCTs does not provide any threshold for grading the bias as either low, high, moderate or others. We had rather presented the overall appraisal in the quality assessment table. To check for the need to add sensitivity analysis for risk of bias (removing high ROB studies), we have added the “Quality” column in the quality assessment table for grading the studies as good, fair or poor as mentioned above (4), and upon assessing the scores, found that no studies were of poor quality (please see S4 Table in the appendix). 

We also appreciate the reviewer’s comment that changes of outliers to the meta-analysis is difficult to interpret. However, outlier analysis was an important step in our meta-analysis, especially as high heterogeneity was observed. Outliers were defined as studies that shift the direction of the effect and/or the heterogeneity of the meta-analysis. The influence of some outliers on the overall meta-analyses was observed, defined as changes in the heterogeneity. However, given that they did not impact the significance or direction of the effect, a decision was made to keep the outliers in the final meta-analysis (6). 

We also agree with the reviewer’s comment that sensitivity/subgroup analysis based on risk of bias and correlation coefficient adds to the robustness of the analysis and findings. Therefore, following the reviewer’s suggestion (feedback 10), subgroup analyses based on quality of methodology of included studies were conducted and reported in S6 Table. Please refer to the authors’ response for feedback 10 for more information. We have also updated the methods section to include the following information:

“Subgroup analyses were performed to investigate differences in the effect of behavior change intervention on QOL scores based on intervention type (DSME, diet and/or physical activity interventions), and settings (hospital/diabetes clinic, community health center, home/web-based), intervention duration (less than 12 weeks, 12 weeks or longer), QOL scale (generic, disease-specific), and methodology quality (good, fair, poor).” (please see page 10, lines 201-205) 

“Meta-regression analyses were performed to determine the linear relationship between duration as a continuous covariate and the effect size.” (please see page 10, lines 205-207) 

Also, meta-analysis of mean QOL was performed using correlation coefficient 0.4 and 0.8 and compared to the assumed coefficient of 0.6. These analyses resulted in changes in magnitude of overall meta-analysis but did not change the direction of the effect or heterogeneity. 

r = 0.4: SMD 1.44 (0.59 to 2.28), p-value=0.0008, I2=96%

r = 0.8: SMD 2.26 (1.01 to 3.50), p-value=0.0004, I2=97.3%

These analyses are provided in the appendix (S6 Table) and explained in the Methods section (page 10 line 207-209).

“Meta analyses using different correlation coefficients (r =0.8 and 0.4) were also conducted to compare the direction of effect and heterogeneity with the assumed coefficient (r = 0.6).” 

and Results (page 30 lines 288-289)

“Meta-analysis using different correlation coefficient (r=0.4 and 0.8) did not change the direction of the effect or the heterogeneity (S6 Table).”

9. Please describe in the results section why there were 14 studies excluded from the meta-analysis, presumably because only subscales of QOL tools were reported? If there were other reasons such as missing variance measures, these can be dealt with by imputing values from other studies and running sensitivity analysis.

Thank you for the comment. Fourteen studies were excluded from the meta-analysis because they only reported the domain specific QOL scores but did not report on the total mean QOL score or composite (physical component summary or mental component summary) scores. We have stated this in the main text (please see page 34 lines 369-370). We have also stated this as one of the limitations of our study (please see page 37 line 452-453). 

10. Interpretation: The findings from the subgroup analysis of delivery setting are heavily overstated in the abstract and main text conclusions. In the abstract conclusions, the authors should likely state that “may substantially improve” or similar, to impart the effect size, rather than just mentioning “significance”. It is unfortunate that the authors did not assess certainty of the evidence (using GRADE or similar) which may have changed some of their conclusions (e.g. “borderline” significant findings for the PCS may have been interpreted differently i.e. possibly a moderate effect size with limitations/reduced certainty due to imprecision and/or inconsistency [causing the wide CI]; risk of bias more integrated into their interpretations if indeed this made any difference to findings). If this is not possible, adding a couple sensitivity (e.g. risk of bias) and subgroup (eg type of QOL scale) may allow for more comment on the robustness of the results and discussion on the specific limitations. Systematic reviews usually use duplicate full text screening and the lack of this should be mentioned in the limitations section.

Thank you for the comment. We acknowledge that the results from the subgroup analysis of delivery setting were overstated in the abstract and main text conclusion. We have removed the emphasis on the subgroup analysis and rephrased the results section in the abstract which now read as follows:

“Of 6122 studies identified initially, 45 studies met the inclusion criteria (n=8336). Of them, 31 involved diabetes self-management education and 14 included dietary and/or physical activity intervention. There was moderate quality evidence from the meta-analysis of mean QOL (n=25) that health behavior interventions significantly improved the QOL of people with T2DM (SMD=1.62, 95%CI=0.65-2.60 I2=0.96, p=0.001). However, no significant improvements were found for studies (n=7) separately assessing the physical component summary (SMD=0.76, 95%CI= -0.03-1.56 I2=0.94, p=0.060) and mental component summary (SMD=0.43, 95%CI= -0.30-1.16 I2=0.94, p=0.249) scores. High heterogeneity and imprecise results across studies resulted in low to moderate quality of evidence.” (please see page 2-3 lines 31-39)

We have also edited the main text conclusion (please see page 38 lines 463-468) and emphasized the low to moderate quality of evidence limiting our conclusion along with the significant finding of the sub-group analysis by intervention setting. The statements read as follows:

“In conclusion, this systematic review and meta-analysis identified health behavior interventions targeted at improving T2DM management as effective strategies in improving the QOL of people with T2DM, with interventions based in community health centres likely to be more effective in improving the physical and mental health component of QOL than those based in clinical or home/web-based settings. However, the low to moderate quality of evidence limits the strength of our conclusion from the pooled evidence.”

We have changed “may significantly improve” to “may substantially improve” in the abstract conclusion (please see page 3 lines 41-42). 

As stated in above responses, we have assessed the certainty of evidence across studies included in the meta-a using the GRADE approach (please see S5 Table in the appendix). According to the table, the certainty of evidence of studies included in the meta-a of mean QOL is of moderate quality whereas, the certainty of evidence of studies included in the PCS and MCS meta-a are of low quality. We have added the findings from the GRADE assessment in the results section (as stated in the above response for feedback 6) and in the conclusion (please see page 38 lines 463-468). The revised sentences in the conclusion read as follows: 

“In conclusion, this systematic review and meta-analysis identified health behavior interventions targeted at improving T2DM management as effective strategies in improving the QOL of people with T2DM, with interventions based in community health centres likely to be more effective in improving the physical and mental health component of QOL than those based in clinical or home/web-based settings. However, the low to moderate quality of evidence limits the strength of our conclusion from the pooled evidence.”

We thank the reviewer for their notes on the full-text screening. Full texts of retrieved studies were screened by two reviewers (AK and PD) independently. We apologise for the writing mistake in the main text and confirm that we have corrected the sentence (please see page 7 lines 139-140), which now reads as follows:

“Full texts of retrieved studies were then screened by two reviewers (AK and PD) independently.”

We appreciate the reviewer’s comment and agree that a subgroup analysis based on methodology quality of the studies will add to the robustness of the findings. The following sections are added to the results:

 “Subgroup analysis based on methodology quality of included studies resulted in a significant improvement in mean QOL in interventions that deemed to have a ‘good’ quality of methodology (n=21). Interventions with ‘fair’ method quality (n=6) did not result in a significant improvement in mean QOL (n=6). However, the test for subgroup analysis was not significant (Q=0.66, p-value=0.41) (S6 Table).” (please see page 31 lines 304-308)

“Subgroup analysis of overall PCS scores based on methodology quality of included studies was not different between studies that deemed to have a ‘good’ methodology quality (n=7) compared to the one study that had a fair methodology quality (S6 Table).” (please see page 32 lines 331-334)

“Subgroup analysis of overall MCS scores based on methodology quality of included studies was not different between studies that deemed to have a ‘good’ methodology quality (n=7) compared to the one study that had a fair methodology quality (S6 Table).” (please see page 33 lines 356-359)

Reviewer #2: Thank you authors for doing a great job on the manuscript. Kindly respond to comments made especially on the methodology section. Again, specify which systematic review methodology you followed. Is it according to JBI format or ?

Thank you for the comment. We have responded to all comments made by the reviewers. We are grateful to have received such constructive feedback and sincerely believe this feedback have greatly helped in improving the quality of the manuscript. We followed the Joanna Briggs Institute (JBI) systematic review methodology and have stated this in the manuscript too (please see page 6 lines 96-98). The statement in the manuscript reads as follows: 

“This systematic review adhered to the Preferred Reporting Items for Systematic Reviews and Meta-Analysis (PRISMA) guideline [21] and followed the Joanna Briggs Institute (JBI) systematic review methodology [22].”

Abstract

Background

 Limited research has assessed this effect in low and middle-income countries (LMICs), nor has it been systematically reviewed. Remember, the research on this area is available but the synthesis of the research done to this effect is limited.

Thank you for your thoughtful comment. We have now revised the sentence and it reads as follows:

“Limited reviews have synthesized this effect in low- and middle-income countries (LMICs)” (please see page 2 lines 18-19)

Methods

• Was there a timeline for this review?

• Which methodology did you follow? Is it the JBI systematic review methodology? 

Thank you for the comment. We started the review with preliminary literature search and review in April 2022; developed the review protocol and submitted to PROSPERO for registration; conducted the systematic search and review using COVIDENCE; synthesised the evidence; reported the findings and prepared a manuscript for publication. Overall, from formulating the research question of the review to submitting the manuscript for publication, the systematic review and meta-analysis was completed in 11 months. Due to abstract word limit, we could not add this information to the abstract, however, we have included this information in the manuscript. 

 “Electronic databases PUBMED/MEDLINE, SCOPUS, CINAHL, Embase, Web of Science and PsycINFO were searched from May to June 2022” (please see page 2 lines 23-24).

“Data were searched from 26 May to 1 June 2022.” (page 6 line 103)

“The articles retrieved from the search were exported to COVIDENCE [24], a web-based software developed to streamline systematic review process on 2 June 2022.” (page 7 lines 136-137)

“The study selection was completed on 22 July 2022.” (page 8 lines 142-143)

“Author name, publication year, study objective, country, setting, study design, demographic characteristics such as age, gender, etc., intervention description, control details, duration of intervention and follow up, QOL measurement tools, and primary and secondary intervention outcomes of interest were extracted using a template by one author (AK) and their accuracy were checked by the second author (PD) [25] from 23 July to 25 August 2022.” (page 8 lines 152-157)

We followed the JBI systematic review methodology and have stated it in the manuscript too (please see page 6 lines 96-98). We couldn’t add this to the abstract due to abstract word limit. The statement in the manuscript reads as follows: 

“This systematic review adhered to the Preferred Reporting Items for Systematic Reviews and Meta-Analysis (PRISMA) guideline [21] and followed the Joanna Briggs Institute (JBI) systematic review methodology [22].”

Conclusion

• What recommendation can be given based on your study results? 

Based on our study results, health behaviour interventions targeted at improving T2DM management may improve quality of life in people with T2DM, however, the interpretation of our findings warrants caution due to the variability of scales used in QOL measurement and low to moderate quality of evidence generated. Therefore, further studies with standardized scales are recommended for more robust estimates. Furthermore, large and well-powered RCTs of high methodological quality are recommended to establish the effect of health behavior interventions on QOL. We have updated our conclusion in the abstract which now reads as follows:

“The findings suggest that health behavior interventions to manage T2DM may substantially improve the QOL of individuals with T2DM. However, due to low to moderate quality of evidence, further research is required to corroborate our findings. Results of this review may guide future research and have policy implications for T2DM management in LMICs.” (page 3 lines 41-44)

We have also updated the recommendation and conclusion in the main text, which reads as follows (please see page 38, lines 463-474):

“In conclusion, this systematic review and meta-analysis identified health behavior interventions targeted at improving T2DM management as effective strategies in improving the QOL of people with T2DM, with interventions based in community health centres likely to be more effective in improving the physical and mental health component of QOL than those based in clinical or home/web-based settings. However, the low to moderate quality of evidence limits the strength of our conclusion from the pooled evidence. Due to the variability of the scales used in QOL measurement and, the interpretation of our findings warrants caution. Further studies with specific standardized scales are recommended for more robust estimates. Furthermore, this review highlights the need for more well-designed RCTs of high methodological quality focusing on QOL as a primary outcome measure. The evidence generated from this review may guide future research and clinical practice and may derive policy implications for the management of T2DM in LMICs.”

Key words

• Rather say health behaviour intervention

• Randomised control trial is not a key word, kindly remove it

Thank you for the suggestion. Following the journal guideline, we have not added any keywords in the manuscript. However, in the online submission system, we have included “health behaviour intervention” and removed “randomised control trial” as keywords. 

Introduction

Remember, this is not primary research. You are looking at synthesis of the available studies whether they are few or not. Therefore, your motivation for the review should not be based on this.

Thank you for the comment. I agree with the reviewer’s suggestion and have made necessary revision. The sentence now reads as follows:

“In sum, studies worldwide have shown that health behavior interventions are effective in optimising glycemic control and diabetes management [12, 19]. However, despite QOL being an important measure in T2DM management, the synthesis of evidence on the effect of these interventions on the QOL of people with T2DM in LMICs is limited.” (please see page 5 lines 85-88)

Methodology

Data sources and searches

• This systematic review adhered to the Preferred Reporting Items for Systematic

 Reviews and Meta-Analysis (PRISMA) guideline [20]. The review protocol was

 registered in the PROSPERO International Prospective Register of systematic reviews

 (Registration ID 90 CRD42022323184). Give a subheading to this section as Reporting and registration of the review. Also give a date when the protocol was registered.

Thank you for the comment. We have added the subheading “Reporting and registration of the review” (please see page 6 line 95). We have also added the date of protocol registration as 5 July 2022 (please see page 6 line 100)

• Kindly add the date for the data searches, how long did it last?

• Online databases PUBMED/MEDLINE, SCOPUS, CINAHL, Embase, Web of Science and

 PsycINFO were searched using the pre-defined search terms. How did you come out with the predefined search terms? Did you use PICO/PIO?

• A manual search of reference lists of included studies was also conducted to identify

 additional studies that met the inclusion criteria. What determined your inclusion and exclusion criteria? This needs to be covered maybe separately so that it becomes clearer to the readers.

Thank you for the comment. Data were searched from 26 May to 1 June 2022. The search terms were built using PICOS framework, which we have added to the main text as well. The revised sentences now read as follows:

“Data were searched from 26 May to 1 June 2022. Online databases PUBMED/MEDLINE, SCOPUS, CINAHL, Embase, Web of Science and PsycINFO were searched using the pre-defined search terms, which were based on the PICOS (population, intervention, comparison, outcome, and study design) framework. The population group was people with T2DM; intervention was any health behavioural intervention targeted at T2DM management; comparison group was usual care, waitlist or attentional control group; and study design was randomized controlled trial.” (please see page 6 lines 103-109)

Our inclusion and exclusion criteria are reported separately under the sub-heading “Inclusion and exclusion criteria” as per the reviewer’s suggestion (please see page 7, line 119-134)

Then data was extracted only from the RCTs conducted in LMICs. Why are you talking of data extraction at this stage of searching in data bases? Whatever you search in data bases is taken to a systematic review app (either covidence or Ryann) before you even start of selection and extraction.

• How did the authors define the LMIC? How many countries make up the LMIC?

Thank you for the comment. We have removed the sentence “The data was extracted only from the RCTs conducted in LMICs” from the “Data sources and searches” paragraph and included the sentence in the “Data extraction” paragraph (please see page 8 line 152).

We defined LMIC adopting the definition of the World Bank (7) and have included this information in the main text (please see page 7 lines 129-130). According to the World Bank, “For the current fiscal year 2023, low-income economies are defined as those having a Gross national income per capita of $1085 or less in 2021; lower middle-income economies are those with a GNI per capita between $1086 and $4255; and upper middle-income economies are those with a GNI per capita between $4256 and $13,205”. Altogether, 136 countries make up the LMIC.

Study selection

•Only studies with an RCT design were included in this review to ensure that the

evidence 103 provided on the effectiveness of interventions was robust. When

multiple articles from the 104 same study/studies were found, only the article that

reported the most relevant QOL 105 information/outcome was included. All types of

health behavior interventions were included 106 regardless of the setting. Only studies

conducted between January 2000 and May 2022 and published in English language

were included in the review, as examining QOL in individuals with T2DM is a relatively

new concept. Studies were included if they 1) examined any behavioral or educational

interventions targeted at improving T2DM management among people with new or

established diagnosis of T2DM; 2) reported QOL outcomes using validated 111 QOL

measures, both pre- and post-intervention, as a primary or secondary outcome; 3)

were conducted within an RCT design; and 4) were conducted in LMICs [21]. Studies

were excluded if they 1) were published in languages other than English; 2) had

therapeutic or pharmacological intervention strategies; 3) had mixed study population

of Type 1 and 2 diabetes with no separate data reported for T2DM population; 4) were

observational studies or reported in reviews, editorials, theses, books, short

communication; or 5) presented inadequate or unclear QOL data. This is the inclusion

and exclusion criteria which should be presented separately.

Thank you for the comment. We have reported our inclusion and exclusion criteria under a separate heading “Inclusion and exclusion criteria” (please see page 7, lines 118-134).

• Selection of articles should start after all the selected articles from the data bases are in Covidence. Once in Covidence, you start with duplicate removal. After duplicate removal, outline the selection based on abstract and title first, then full text screening.

Thank you for the comment. We have moved the heading “Study selection” further down in the manuscript which now contains information as suggested by the reviewer, that is, export of retrieved articles to COVIDENCE, duplicates removal, title and abstract screening and full-text screening (please see page 7-8, lines 135-143).

• Full texts of retrieved studies were then screened by the first reviewer (AK). Why the full text screening done by one reviewer? Because two reviewers are supposed to do independently with blind on, then solve the disagreements later.

Thank you for pointing this out. We apologise for the writing mistake in the main text. Full texts of retrieved studies were screened by two reviewers (AK and PD) independently. Both title, abstract and full text were screened by two reviewers (AK and PD) independently. We have corrected the sentence in the manuscript which now reads as follows:

“Full texts of retrieved studies were then screened by two reviewers (AK and PD) independently.” (please see page 7 lines 139-140)

• Any disagreements were resolved through consultation with other team members. What type of disagreements were resolved or were there any?

Thank you for the comment. The following type of disagreements were resolved through consensus among team members. There was a disagreement between two authors (AK and PD) on whether to keep RCTs evaluating a mix of pharmacological and non-pharmacological (e.g. health behaviours) intervention (8) or multi arm trials with both health behaviour intervention and pharmacological intervention (9). Disagreements were then resolved through consultation with other team members and a decision was made to exclude those RCTs that had a mix of health behaviour and pharmacological intervention (as it was difficult to ascertain the intervention effect in multi arm trials).

• Add the selection dates, how long did the selection take?

The selection dates are from 2 June 2022 (import of references to COVIDENCE) to 22 July 2022 (completion of full text screening). Hence, the study selection took seven weeks. We have added the study selection dates in the manuscript (please see page 7, line 137 and page 8, line 143).

Data Extraction

• Kindly add the dates for the data extraction

Thank you for the suggestion. The dates of the data extraction are from 23 July 2022 to 25 August 2022. These dates are added in the manuscript too (please see page 8, line 157).

References

1. Didsbury MS, Kim S, Medway MM, Tong A, McTaggart SJ, Walker AM, et al. Socio-economic status and quality of life in children with chronic disease: A systematic review. J Paediatr Child Health. 2016;52(12):1062-9.

2. Fielding S, Ogbuagu A, Sivasubramaniam S, MacLennan G, Ramsay CR. Reporting and dealing with missing quality of life data in RCTs: has the picture changed in the last decade? Qual Life Res. 2016;25(12):2977-83.

3. Younge JO, Kouwenhoven-Pasmooij TA, Freak-Poli R, Roos-Hesselink JW, Hunink MM. Randomized study designs for lifestyle interventions: a tutorial. International Journal of Epidemiology. 2015;44(6):2006-19.

4. John JR, Jani H, Peters K, Agho K, Tannous WK. The Effectiveness of Patient-Centred Medical Home-Based Models of Care versus Standard Primary Care in Chronic Disease Management: A Systematic Review and Meta-Analysis of Randomised and Non-Randomised Controlled Trials. Int J Environ Res Public Health. 2020;17(18).

5. Joanna Briggs Institute. Critical Appraisal Tools 2017 [Available from: https://jbi.global/critical-appraisal-tools.

6. Baker R, Jackson D. A new approach to outliers in meta-analysis. Health Care Manag Sci. 2008;11(2):121-31.

7. The World Bank. World Bank Country and Lending Groups 2022 [Available from: https://datahelpdesk.worldbank.org/knowledgebase/articles/906519-world-bank-country-and-lending-groups.

8. Tu Q, Xiao LD, Ullah S, Fuller J, Du H. A transitional care intervention for hypertension control for older people with diabetes: A cluster randomized controlled trial. J Adv Nurs. 2020;76(10):2696-708.

9. Halperin F, Ding SA, Simonson DC, Panosian J, Goebel-Fabbri A, Wewalka M, et al. Roux-en-Y gastric bypass surgery or lifestyle with intensive medical management in patients with type 2 diabetes: feasibility and 1-year results of a randomized clinical trial. JAMA Surg. 2014;149(7):716-26.

---

## [Decision Letter · Decision Letter 1]

15 Aug 2023

PONE-D-23-05175R1The effect of health behavior interventions to manage Type 2 diabetes on the quality of life in low-and middle-income countries: a systematic review and meta-analysisPLOS ONE

Dear Dr. Karki,

Thank you for submitting your manuscript to PLOS ONE. After careful consideration, we feel that it has merit but does not fully meet PLOS ONE’s publication criteria as it currently stands. Therefore, we invite you to submit a revised version of the manuscript that addresses the points raised during the review process.

As you will see from Reviewer 1's comments, there is room to improve the revised version of the manuscript to ensure that conclusions drawn from the paper closely reflect the data.

We look forward to receiving your revised manuscript.

Kind regards,

Edward Zimbudzi

Academic Editor

PLOS ONE

Journal Requirements:

Reviewers' comments:

Reviewer's Responses to Questions

**Comments to the Author**

1. If the authors have adequately addressed your comments raised in a previous round of review and you feel that this manuscript is now acceptable for publication, you may indicate that here to bypass the “Comments to the Author” section, enter your conflict of interest statement in the “Confidential to Editor” section, and submit your "Accept" recommendation.

Reviewer #1: (No Response)

Reviewer #2: All comments have been addressed

2. Is the manuscript technically sound, and do the data support the conclusions?

Reviewer #1: Partly

Reviewer #2: Yes

3. Has the statistical analysis been performed appropriately and rigorously? 

Reviewer #1: Yes

Reviewer #2: Yes

4. Have the authors made all data underlying the findings in their manuscript fully available?

Reviewer #1: Yes

Reviewer #2: Yes

5. Is the manuscript presented in an intelligible fashion and written in standard English?

Reviewer #1: Yes

Reviewer #2: Yes

6. Review Comments to the Author

Reviewer #1: The authors have addressed many of the suggestions from the previous peer review. I still believe that there can be more improvements to ensure current review methodology is followed and that the conclusions about the findings closely reflect the data.

Major comments:

1. I strongly object to using lack of feasibility of blinding as a reason to not rate the studies at risk of bias from lack of blinding. There is still the risk of bias regardless of blinding not being practical. This is commonly mentioned at training workshops to avoid doing. The authors may not think the risk is that serious to potentially rate down during GRADE, but the risk of bias from lack of blinding does exist.

2. The main text conclusions paragraph still mentions that the community setting is "likely" an impact on PCS and MCS findings, and this should be downplayed substantially.

3. Now that the authors have clarified that their results represent the immediate effects after the interventions, they should add the caveat that their findings are for the short-term QOL of participants and that that any lasting/sustained effect was not examined in this review. I think the term "follow-up effect" (not too sure if that is a used term) should be removed with focus that the authors chose to keep things consistent across studies using a similar timepoint.

Minor:

abstract: suggest to remove "significant" in the sentence about the mean QOL findings.

methods: delete the first sentence in the data extraction section (this refers to eligibility criteria); was <=5 or <=6 used for GRADE study limitations domain? This seems inconsistent with the risk of bias categories; please mention that the added subgroups for study quality and measurement tool were added post hoc (after protocol).

results: are the findings for the overall QOL analysis SMD 1.46 or 1.62? this has changed since the original submission and is not consistent throughout the paper.

I had hoped that the authors would have removed the "influencer" and outlier analyses, which are not current practice due to their focusing on results rather than clinical and methodological differences between studies, and would encourage them to reconsider this. The fact that these analyses did not impact the findings is not a good reason to keep them in the paper. We do not just remove/exclude a study because of it's findings being "different" from others, unless there is a methodological or clinical reason to do so. What if they study was actually the best conducted and largest??

Lastly I think the authors could consider focusing on the (moderate certainty) overall QOL findings and speak to these a bit more strongly as indicating benefit at least over the short term. One recommendation may be for more longer term follow-up to see if the effects are sustainable.

Thank you

Reviewer #2: Thank you authors for addressing the comments raised. Congratulations on the job well done and this has improved the quality of the manuscript.

7. PLOS authors have the option to publish the peer review history of their article (what does this mean?). If published, this will include your full peer review and any attached files.

Reviewer #1: No

Reviewer #2: No

---

## [Author Response · Author response to Decision Letter 1]

27 Aug 2023

The Editor-in-Chief

PLOS One

Subject: Submission of revised manuscript on “The effect of health behavior interventions to manage Type 2 diabetes on the quality of life in low-and middle-income countries: a systematic review and meta-analysis”

Thank you for providing us with the opportunity to submit a revised manuscript “The effect of health behavior interventions to manage Type 2 diabetes on the quality of life in low-and middle-income countries: a systematic review and meta-analysis” for consideration of publication in PLOS One.

We express our gratitude to the reviewers for providing feedback and their insightful comments on the manuscript, which certainly have helped to enhance the quality of the manuscript. We would like to confirm that the reference list is complete and correct and no retracted papers have been cited. 

Please find a point-by-point response to the editor’s and reviewers’ comments below. Author’s responses are marked blue. All page and line numbers refer to the track changed version of the revised manuscript.

Reviewer #1: The authors have addressed many of the suggestions from the previous peer review. I still believe that there can be more improvements to ensure current review methodology is followed and that the conclusions about the findings closely reflect the data.

Thank you for your thoughtful comment. We have addressed all your comments. 

Major comments:

1. I strongly object to using lack of feasibility of blinding as a reason to not rate the studies at risk of bias from lack of blinding. There is still the risk of bias regardless of blinding not being practical. This is commonly mentioned at training workshops to avoid doing. The authors may not think the risk is that serious to potentially rate down during GRADE, but the risk of bias from lack of blinding does exist.

Thank you for the comment. Upon further consideration and discussion with the co-authors, we agree with the reviewer and acknowledge that the lack of blinding poses a risk of bias in behavioral interventions regardless of whether blinding was feasible or not. Hence, studies not reporting blinding of either the participants, personnel or outcome assessors were given a score of “0” and marked as “N” for “No” under the respective items about blinding of participants, personnel or outcome assessors in the 13-item Joanna Briggs Institute (JBI) Critical Appraisal Checklist for Randomized Controlled Trials. We have detailed out all this information in the Quality Assessment section under Methods and results both, as following (please see pages 8-9, lines 160-180):

In Methods:

“The 13-item Joanna Briggs Institute (JBI) Critical Appraisal Checklist for Randomized Controlled Trials was used to assess methodological quality of the selected studies [26]. Two reviewers (AK and PD) independently assessed the risk of bias. Any disagreements were discussed with the team members until consensus was reached. Each item in the JBI checklist was scored one if they fulfilled the criteria for that item and scored zero if they did not fulfil the criteria. For example, if a study reported blinding of outcome assessors, then the study was scored ‘1’ for the item about blinding of outcome assessors, whereas, if the study did not report blinding of outcome assessors, then the study was scored ‘0’. Summary scores were obtained for each selected studies by adding the item-specific scores. The quality of the studies was then rated as good (≥8), fair (6-7), or poor (≤5) based on the summary scores [27]. In addition, the quality of evidence across RCTs included in the meta-analysis was assessed by two reviewers (AK and PD) using the Grading of Recommendations, Assessment, Development and Evaluation (GRADE) approach, and rated as high, moderate, low or very low [28]. Since all included studies were RCTs, the rating began with a high-certainty rating. Then, the quality was upgraded or downgraded based on the following criteria: i) quality across the studies as determined by the JBI Critical Appraisal Checklist for Randomized Controlled Trials; ii) inconsistency/heterogeneity level; iii) indirectness of evidence; iv) imprecise results (wide 95% CI i.e., > 0.8 SMD); and v) publication bias (visual inspection of funnel plot) [29]. If the majority of the RCTs scored ≤ 5 on the JBI scale, the certainty of evidence was downgraded two places whereas, if the effect size was large (SMD ≥ 0.8), then the evidence was upgraded one place [29].”

From the Results section, we have removed the statement “Blinding of participants is generally difficult in behavioral interventions due to participants being aware whether they receive the intervention, hence studies that did not address or clarify blinding were not downgraded for those risks or uncertainties.” (page 31, lines 283-286). We have also added a description in the Results section, which read as follows: 

“The overall quality of the studies ranged from “fair” to “good” after obtaining a summary score of at least six in the quality assessment. No study was rated as poor quality as none obtained a summary score of five or less.” (please see page 31, lines 278-280) 

2. The main text conclusions paragraph still mentions that the community setting is "likely" an impact on PCS and MCS findings, and this should be downplayed substantially.

We thank the reviewer for noting this. We have removed the statement about community setting having an impact on the PCS and MCS findings. The removed statement is in page 39 lines 480-482 of the track changed document. 

3. Now that the authors have clarified that their results represent the immediate effects after the interventions, they should add the caveat that their findings are for the short-term QOL of participants and that that any lasting/sustained effect was not examined in this review. I think the term "follow-up effect" (not too sure if that is a used term) should be removed with focus that the authors chose to keep things consistent across studies using a similar timepoint.

Thank you for this comment. Since we measured the immediate post-intervention effects of the interventions and didn’t measure the longer-term effect of the interventions, we have added it as one of the limitations of the study in page 38-39 lines 466-469. The statement now reads as follows:

“Thirdly, the findings of this meta-analysis only related to short-term QOL (i.e., immediate post-intervention effects) and any lasting/sustained effect was not examined in this review as there was no sufficient data to examine longer-term effects on QOL.”

We have removed the term “follow-up effect” and revised the statement, which now reads as follows (page 10 lines 202-206):

“For studies with QOL measurements at multiple time-points, only the measurements at baseline and following intervention completion (post-intervention) were included in the meta-analysis to maintain consistency across studies using a similar timepoint and improve homogeneity in data extraction.”

Minor:

abstract: suggest to remove "significant" in the sentence about the mean QOL findings.

Thank you for the comment. The word “significant” has been removed from the statement, which now reads as below (please see page 2, line 34):

“There was moderate quality evidence from the meta-analysis of mean QOL (n=25) that health behavior intervention improved the QOL of people with T2DM (SMD=1.62, 95%CI=0.65-2.60 I2=0.96, p=0.001).”

methods: delete the first sentence in the data extraction section (this refers to eligibility criteria); was <=5 or <=6 used for GRADE study limitations domain? This seems inconsistent with the risk of bias categories; please mention that the added subgroups for study quality and measurement tool were added post hoc (after protocol).

Thank you for the comment. The first sentence from the data extraction section in the manuscript body, which initially read as “Data was extracted only from the RCTs conducted in LMICs” has been removed. (see page 8, line 153)

≤5 was used as the GRADE study limitation domain to be consistent with the risk of bias categories. We have corrected the inconsistencies and replaced ≤6 with ≤5. The sentence now reads as below (please see page 9, lines 177-180):

“If the majority of the RCTs scored ≤5 on the JBI scale, the certainty of evidence was downgraded two places whereas, if the effect size was large (SMD ≥ 0.8), then the evidence was upgraded one place [29].” 

We have added a statement about performing post-hoc subgroup analyses based on study quality and QOL scale, which reads as follows: 

“Sub-group analyses based on QOL scale and methodology quality were added post hoc.” (see page 10 lines 213-214)

results: are the findings for the overall QOL analysis SMD 1.46 or 1.62? this has changed since the original submission and is not consistent throughout the paper.

Thank you for the comment. The correct SMD is 1.62. We have made necessary correction at page 32-33 line 323-324). The corrected sentence now reads as follows:

 “However, due to large effect size (SMD: 1.62), the studies were upgraded by one level.”

I had hoped that the authors would have removed the "influencer" and outlier analyses, which are not current practice due to their focusing on results rather than clinical and methodological differences between studies, and would encourage them to reconsider this. The fact that these analyses did not impact the findings is not a good reason to keep them in the paper. We do not just remove/exclude a study because of it's findings being "different" from others, unless there is a methodological or clinical reason to do so. What if they study was actually the best conducted and largest??

Thank you for the comment. We have removed the “influence” and “outlier” analyses as suggested by the reviewer. We have removed all the following texts in relation to influence and outlier analyses from the manuscript:

“Outliers were identified statistically and graphically.” (page 10, line 199)

“Sensitivity analyses were performed using a leave-one-out method, where individual trials are excluded one at a time and the changes in overall results and heterogeneity are explored.” (page 10, lines 207-208)

“Influence analyses (S2 Fig) suggested that the overall meta-analysis was sensitive to the study by Safavi et al (74). Excluding this study did not change the direction of the effect (SMD: 1.17, 95% CI: 0.70 to 1.64, I2: 0.95).” (page 32, lines 303-305)

“Outlier analysis identified nine studies as outliers [8, 46, 48, 50, 52, 55, 61, 63, 74]. Excluding these studies reduced the heterogeneity slightly (to 90.1%) and the magnitude but not the overall direction of the effect (SMD: 1.14, 95% CI: 0.83 to 1.45, p-value<0.001).” (page 33, lines 326-328)

“Influence analyses (S4 Fig) suggested that the overall meta-analysis was sensitive to the study by Rias et al. [49]. Excluding this study did not change the direction or the significance of the effect (SMD: 0.458, 95% CI: -0.125 to 1.040, I2: 0.922).” (page 33, lines 336-338)

“Outlier analysis identified two studies as outliers [44, 49]. Excluding these studies reduced the heterogeneity to 0% and resulted in an overall significant effect (SMD 0.18, 95% CI: 0.03 to 0.34, p-value<0.05).” (page 34, lines 352-354)

“Influence analyses (S6 Fig) suggested that the overall meta-analysis was sensitive to the study by Sekhar et al. [44]. However, excluding this study did not change the direction or the significance of the effect (SMD: 0.094, 95% CI: -0.349 to 0.537, I2: 0.883).” (page 34, lines 361-363)

“Two studies were identified as outliers in the outlier analysis [44, 73]. Excluding these studies reduced the heterogeneity to 72.4% but not the overall direction of the effect (SMD 0.24, 95% CI: -0.17 to 0.65, p-value<0.256).” (page 35, lines 377-379)

We have also removed the supporting information caption of sensitivity analyses, which initially read as below:

 “S2 Fig. Sensitivity analysis of Mean Quality of life” (page 47 line 790)

 “S4 Fig. Sensitivity analysis of Mean Physical Component Summary” (page 47 line 792)

 “S6 Fig. Sensitivity analysis of mean Mental Component Summary” (page 47 line 795)

Lastly, I think the authors could consider focusing on the (moderate certainty) overall QOL findings and speak to these a bit more strongly as indicating benefit at least over the short term. One recommendation may be for more longer term follow-up to see if the effects are sustainable.

Thank you for this comment. We have rephrased the main text conclusion focusing on the moderate certainty of evidence of overall QOL findings indicating benefits over at least short period of time. The revised conclusion reads as follows (please see page 39-40 lines 478-493):

“In conclusion, this systematic review and meta-analysis identified health behavior interventions targeted at improving T2DM management as effective strategies in improving the QOL of people with T2DM. The study demonstrated a moderate certainty of evidence for the overall QOL findings, indicating that the immediate post-intervention improvement on the overall QOL of people with T2DM is likely to be close to the true effect of the interventions on QOL over the short term. However, the low quality of evidence of PCS and MCS findings limits the strength of our conclusion from the pooled evidence. Due to the variability of the scales used in QOL measurement, the interpretation of our findings warrants caution. Studies with specific standardized scales are recommended for more robust estimates. Further research is needed to examine the long-term effectiveness of health behavior interventions on QOL. Furthermore, this review highlights the need for more well-designed RCTs of high methodological quality focusing on QOL as a primary outcome measure. The evidence generated from this review may guide future research and clinical practice and may derive policy implications for the management of T2DM in LMICs.”

We have added a recommendation about the need for conducting further studies assessing the longer-term effects of the interventions; the statement reads as follows (see page 39 lines 489-490):

“Further research is needed to examine the long-term effectiveness of health behaviour interventions on QOL.”

Reviewer #2: Thank you authors for addressing the comments raised. Congratulations on the job well done and this has improved the quality of the manuscript.

Thank you. We appreciate the valuable feedback from the reviewers.

---

## [Decision Letter · Decision Letter 2]

4 Oct 2023

The effect of health behavior interventions to manage Type 2 diabetes on the quality of life in low-and middle-income countries: a systematic review and meta-analysis

PONE-D-23-05175R2

Dear Dr. Karki,

We’re pleased to inform you that your manuscript has been judged scientifically suitable for publication and will be formally accepted for publication once it meets all outstanding technical requirements.

Kind regards,

Edward Zimbudzi

Academic Editor

PLOS ONE

Additional Editor Comments (optional):

Reviewers' comments:

Reviewer's Responses to Questions

**Comments to the Author**

1. If the authors have adequately addressed your comments raised in a previous round of review and you feel that this manuscript is now acceptable for publication, you may indicate that here to bypass the “Comments to the Author” section, enter your conflict of interest statement in the “Confidential to Editor” section, and submit your "Accept" recommendation.

Reviewer #1: All comments have been addressed

2. Is the manuscript technically sound, and do the data support the conclusions?

Reviewer #1: (No Response)

3. Has the statistical analysis been performed appropriately and rigorously? 

Reviewer #1: (No Response)

4. Have the authors made all data underlying the findings in their manuscript fully available?

Reviewer #1: (No Response)

5. Is the manuscript presented in an intelligible fashion and written in standard English?

Reviewer #1: (No Response)

6. Review Comments to the Author

Reviewer #1: (No Response)

7. PLOS authors have the option to publish the peer review history of their article (what does this mean?). If published, this will include your full peer review and any attached files.

Reviewer #1: No

---

## [Editor Report · Acceptance letter]

8 Oct 2023

PONE-D-23-05175R2 

The effect of health behavior interventions to manage Type 2 diabetes on the quality of life in low-and middle-income countries: a systematic review and meta-analysis 

Dear Dr. Karki:

I'm pleased to inform you that your manuscript has been deemed suitable for publication in PLOS ONE. Congratulations! Your manuscript is now with our production department. 

Kind regards, 

on behalf of

Dr. Edward Zimbudzi 

Academic Editor

PLOS ONE